# GraphFLEx: Structure Learning Framework for Large Expanding Graphs

## Abstract

Graph structure learning is a fundamental problem critical for interpretability and uncovering relationships in data. While graphical data is central to information representation, inferring graph structures remains challenging. Existing methods falter with expanding graphs, requiring costly re-learning of the entire structure for new nodes, and face severe computational and memory demands on large graphs. To overcome these challenges, we propose **GraphFLEx**: a unified framework for structure learning in Large and Expanding Graphs. GraphFLEx efficiently limits potential connections to relevant nodes by leveraging clustering and coarsening techniques, significantly reducing computational costs and enhancing scalability. **GraphFLEx** provides 48 flexible methods for graph structure learning by integrating diverse learning, coarsening, and clustering approaches. Extensive experiments with various GNN models demonstrate its effectiveness. Our code is available here.

## 1. Introduction

Graph representations capture relationships between entities, vital across diverse fields like biology, finance, sociology, engineering, and operations research (Zhou et al., 2020; Fout et al., 2017; Wu et al., 2020). While some relationships, such as social connections or sensor networks, are directly observable, many, including gene regulatory networks, scene graph generation (Gu et al., 2019), brain networks, (Zhu et al., 2021) and drug interactions, require inference (Allen et al., 2012). Even when available, graph data often contains noise, requiring denoising and recalibration. Thus, inferring graph structures becomes crucial, often surpassing the choice of graph or algorithm itself.

*Graph Structure Learning (GSL)* offers a solution, enabling the construction and refinement of graph topologies. GSL has been widely studied in both supervised and unsupervised

contexts (Liu et al., 2022; Chen & Wu, 2022). In supervised GSL (s-SGL), the adjacency matrix and Graph Neural Networks (GNNs) are jointly optimized for a downstream task, such as node classification. Notable examples of s-GSL include $NodeFormer$ (Wu et al., 2022), $Pro-GNN$ (Jin et al., 2020), $WSGNN$ (Lao et al., 2022), and $SLAPS$ (Fatemi et al., 2021). Unsupervised GSL (u-SGL), on the other hand, focuses solely on learning the underlying graph structure, typically through adjacency or Laplacian matrices. Methods in this category include approximate nearest neighbours ($A-NN$) (Dong et al., 2011; Muja & Lowe, 2014), k-nearest neighbours ($k-NN$) (MacQueen et al., 1967; Wang & Zhang, 2006), covariance estimation ($emp.Cov.$) (Hsieh et al., 2011), graphical lasso ($GLasso$) (Friedman et al., 2008), and signal processing techniques like $l2$-model,$log$-model, and $large$-model (Dong et al., 2016; Kalofolias, 2016).

While s-SGL methods offer promising results, they have limitations: (1) they rely on label information, restricting their applicability in settings without annotations; (2) they are often task-specific, optimizing for node classification rather than general graph topology (Liu et al., 2022). These issues are avoided in u-SGL approaches, which are the focus of this work. However, both s-SGL and u-SGL face challenges when applied to large-scale or expanding datasets.

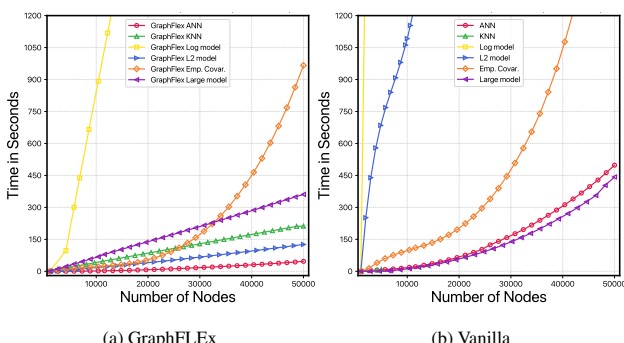

(a) GraphFLEx      (b) Vanilla

Figure 1: High computational time required to learn graph structures using existing methods, whereas GraphFLEx effectively controls computational growth, achieving near-linear scalability. Notably, Vanilla KNN failed to construct graph structures with fewer than 10k nodes due to memory limitations.

As contemporary datasets grow in size, scalability becomes a critical challenge, with existing methods proving too computationally expensive for large-scale graphs. In such cases, Approximate Nearest Neighbours ($A-NN$), with time com-

[1]Anonymous Institution, Anonymous City, Anonymous Region, Anonymous Country. Correspondence to: Anonymous Author <anon.email@domain.com>.

Preliminary work. Under review by the International Conference on Machine Learning (ICML). Do not distribute.

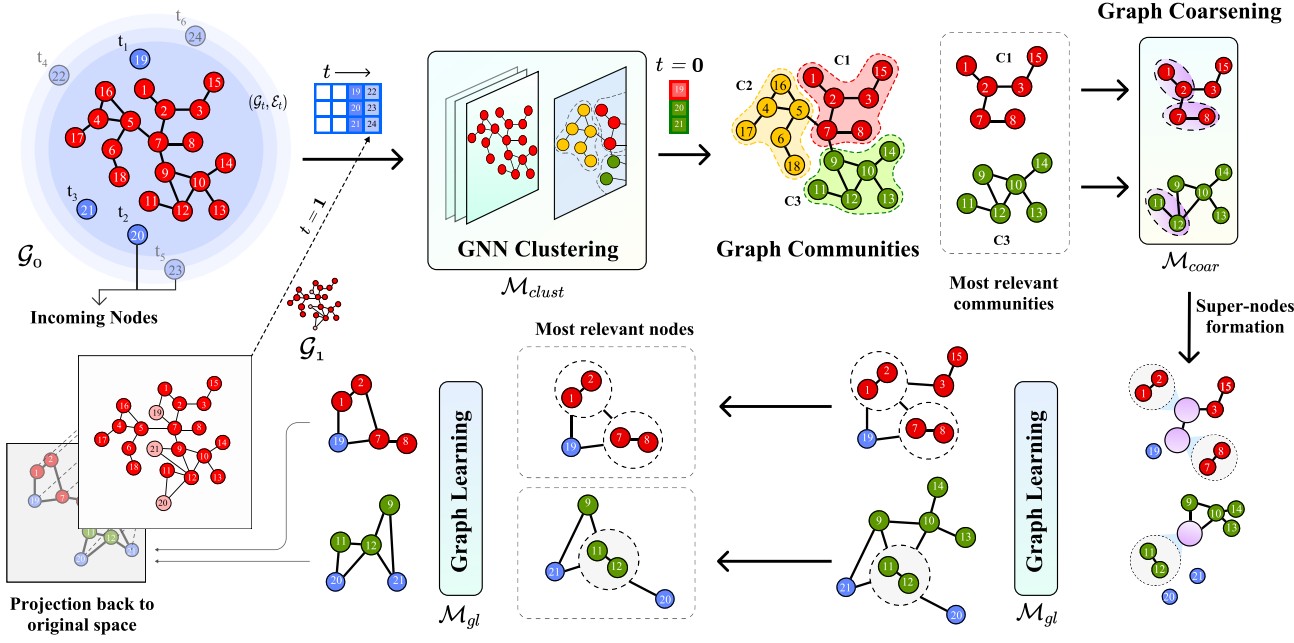

Figure 2: General pipeline of GraphFLEx, it processes a graph ($\mathcal{G}_{t-1}$) and incoming nodes ($\mathcal{E}_t$) at time $t$, comprising three main components: a) **Clustering**, which infers $\mathcal{E}_t$ nodes to existing communities using a pre-trained model $\mathcal{M}_{\text{clust}}(\mathcal{G}_0)$; b) **Coarsening**, reduces the size of the desired community; and c) **Learning**, where the structure associated with $\mathcal{E}_t$ nodes are learned using the coarsened graph, followed by projecting this structure onto the original graph to create graph $\mathcal{G}_t$ at time $t$.

plexity $O(N \log(N))$, is often the only feasible solution. In contrast, methods like $k$-NN, $log$-model, and $l2$-model are significantly more costly, with time complexities exceeding $\mathcal{O}(N^2)$.

The aforementioned techniques are ineffective for learning large-scale graphs because they consider the entire collection of nodes to determine connections for every individual node. All nodes, however, only have connections to a very small set of nodes. Therefore, we need to devise a method that can refine the entire graph's node set to a smaller subset of potential node sets, with the aim of identifying feasible connections. Additionally, expanding graphs where new nodes continuously arrive further complicates the issue, as existing methods require re-learning the entire graph structure with each new node (Khazane et al., 2019; Holme & Saramäki, 2012). This makes them inefficient for expanding data. To address these challenges, we propose GraphFLEx, a comprehensive framework that tackles both scalability for large datasets and adaptability for growing graphs.

As shown in Figure 2, **GraphFLEx** comprises three key modules: (i) Graph Clustering, (ii) Graph Coarsening, and (iii) Graph Learning. By leveraging clustering and coarsening, GraphFLEx significantly reduces computational overhead by restricting possible connections to only relevant nodes. Figure 1 compares the graph structure learning time, highlighting GraphFLEx's efficiency over existing methods. Key contributions of GraphFLEx include:

**Key Contributions and Novelty.**

- We provide strong *theoretical guarantees* that the structure learned from a small subset of nodes is equivalent to that learned from the full set. This is supported by empirical results using real-world and synthetic datasets, demonstrating the effectiveness of GraphFLEx across diverse graph structures.

- GraphFLEx is composed of independently operating modules, allowing the creation of new learning frameworks by modifying any of its three modules. It currently supports *48 distinct methods* for learning graph structure, offering flexibility across various domains.

- GraphFLEx efficiently handles *large-scale and expanding graphs*, enhancing scalability for graph learning tasks.

- GraphFLEx serves as a *comprehensive framework* applicable individually for clustering, coarsening, and learning tasks.

## 2. Problem Formulation and Background

A graph $\mathcal{G}$ is represented using $\mathcal{G}(V, A, X)$ where $V = \{v_1, v_2...v_N\}$ is the set of $N$ nodes, each node $v_i$ has a $d-$dimensional feature vector $x_i$ in $X \in \mathbb{R}^{N \times d}$ and $A \in \mathbb{R}^{N \times N}$ is adjacency matrix representing connection between $i^{th}$ and $j^{th}$ nodes when entry $A_{ij} > 0$. An expanding graph $\mathcal{E}_{\mathcal{G}}$ can be considered a variant of graph $\mathcal{G}$ where nodes $v$ now have an associated timestamp $\tau_v$. We can represent a expanding graph as a sequence of graphs, i.e., $\mathcal{E}_{\mathcal{G}} = \{\mathcal{G}_0, \mathcal{G}_1, ... \mathcal{G}_T\}$ where $\{\mathcal{G}_0 \subseteq \mathcal{G}_1.... \subseteq \mathcal{G}_T\}$ at

$\tau \in \{0, ... T\}$ timestamps. New nodes arriving at different timestamps are seamlessly integrating into initial graph $\mathcal{G}_0$.

**Problem statement.** Given a partially known or missing graph structure, our goal is to incrementally learn the whole graph, i.e., learn adjacency or laplacian matrix. Specifically, we consider two unsupervised GSL tasks:

**Goal 1.** *Large Datasets with Missing Graph Structure: In this setting, the graph structure is entirely unavailable, and existing methods are computationally infeasible for learning the whole graph in a single step. To address this issue, we first randomly partition the dataset into exclusive subsets. We then learn the initial graph $\mathcal{G}_0(V_0, X_0)$ over a small subset of nodes and incrementally expand it by integrating additional partitions, ultimately reconstructing the full graph $\mathcal{G}_T$.*

**Goal 2.** *Partially Available Graph: In this case, we only have access to the graph $\mathcal{G}_t$ at timestamp t, with new nodes arriving over time. The goal is to update the graph incrementally to obtain $\mathcal{G}_T$, without re-learning it from scratch at each timestamp.*

GraphFlex addresses these challenges with a unified framework, outlined in Section 3. Before delving into the framework, we review some key concepts.

### 2.1. Graph Reduction

Graph reduction encompasses sparsification, clustering, coarsening, and condensation (Hashemi et al., 2024). GraphFlex employs clustering and coarsening to refine the set of relevant nodes for potential connections.

**Graph Clustering.** Graphs often exhibit global heterogeneity with localized homogeneity, making them well-suited for clustering (Fortunato, 2010). Clusters capture higher-order structures, aiding graph learning. Methods like DMoN (Tsitsulin et al., 2023) use GNNs for soft cluster assignments, while Spectral Clustering (SC) (Kamvar et al., 2003) and K-means (Wagstaff et al., 2001; MacQueen et al., 1967) efficiently detect communities. DiffPool (Bruna et al., 2014; Defferrard et al., 2016) applies SC for pooling in GNNs.

**Graph Coarsening.** Graph Coarsening (GC) reduces a graph $\mathcal{G}(V, E, X)$ with $N$ nodes and features $X \in \mathbb{R}^{N \times d}$ into a smaller graph $\mathcal{G}_c(\widetilde{V}, \widetilde{E}, \widetilde{X})$ with $n \ll N$ nodes and $\widetilde{X} \in \mathbb{R}^{n \times d}$. This is achieved via learning a coarsening matrix $\mathcal{P} \in \mathbb{R}^{n \times N}$, mapping similar nodes in $\mathcal{G}$ to super-nodes in $\mathcal{G}_c$, ensuring $\widetilde{X} = \mathcal{P}X$ while preserving key properties (Loukas, 2019; Kataria et al., 2023; Kumar et al., 2023; Kataria et al., 2024).

### 2.2. Unsupervised Graph Structure Learning

Unsupervised graph learning spans from simple k-NN weighting (Wang & Zhang, 2006; Zhu et al., 2003) to advanced statistical and graph signal processing (GSP) tech-

Table 1: Unsupervised Graph Structure Learning Methods

| Method | Time Complexity | Formulation |
|---|---|---|
| $GLasso$ | $O(N^3)$ | $\max_\Theta \log \det \Theta$ $-\text{tr}(\hat{\Sigma}\Theta) - \rho\|\Theta\|_1$ |
| $log$-model | $O(N^2)$ | $\min_{W \in \mathcal{W}} \|W \circ Z\|_{1,1}$ $-\alpha\mathbf{1}^T \log(W\mathbf{1}) + \frac{\beta}{2}\|W\|_F^2$ |
| $l2$-model | $O(N^2)$ | $\min_{W \in \mathcal{W}} \|W \circ Z\|_{1,1}$ $+\alpha\|W\mathbf{1}\|^2 + \alpha\|W\|_F^2$ $+\mathbf{1}\{\|W\|_{1,1} = n\}$ |
| $large$-model | $O(N \log(N))$ | $\min_{W \in \tilde{\mathcal{W}}} \|W \circ Z\|_{1,1}$ $-\alpha\mathbf{1}^T \log(W\mathbf{1}) + \frac{\beta}{2}\|W\|_F^2$ |

niques. Statistical methods, also known as probabilistic graphical models, assume an underlying graph $\mathcal{G}$ governs the joint distribution of data $X \in \mathbb{R}^{N \times d}$ (Koller & Friedman, 2009; Banerjee et al., 2008; Friedman et al., 2008). Some approaches (Dempster, 1972) prune elements in the inverse sample covariance matrix $\widehat{\Sigma} = \frac{1}{d-1}XX^T$ and sparse inverse covariance estimators, such as Graphical Lasso (GLasso) (Friedman et al., 2008): $\text{maximize}_\Theta \log \det \Theta - \text{tr}(\widehat{\Sigma}\Theta) - \rho\|\Theta\|_1$, where $\Theta$ is the inverse covariance matrix. However, these methods struggle with small sample sizes. Graph Signal Processing (GSP) techniques analyze signals on known graphs, ensuring properties like smoothness and sparsity. Signal smoothness on a graph $\mathcal{G}$ is quantified by the Laplacian quadratic form:

$$Q(\mathbf{L}) = \mathbf{x}^T\mathbf{L}\mathbf{x} = \frac{1}{2}\sum_{i,j} w_{ij}(\mathbf{x}(i) - \mathbf{x}(j))^2.$$

For a set of vectors $X$, smoothness is measured using the Dirichlet energy (Belkin et al., 2006): $\text{tr}(X^T L X)$. State-of-the-art methods (Dong et al., 2016; Kalofolias, 2016; Hu et al., 2013) optimize Dirichlet energy while enforcing sparsity or specific structural constraints. Table 1 compares various graph learning methods based on their formulations and time complexities.

*Remark* 1. Graph Structure Learning (GSL) differs significantly from Continual Learning (CL) (Van de Ven & Tolias, 2019; Zhang et al., 2022; Parisi et al., 2019) and Dynamic Graph Learning (DGL) (Kim et al., 2022; Wu et al., 2023; You et al., 2022), as discussed in Appendix C.

## 3. GraphFLEx

In this section, we introduce GraphFLEx, which has three main modules:

- **Graph Clustering.** Identifies communities and extracts higher-order structural information,
- **Graph Coarsening.** Is used to coarsen down the desired community, if the community itself is large,
- **Graph Learning.** Learns the graph's structure using a

limited subset of nodes from the clustering and coarsening modules, *enabling scalability*.

For more details, see Algorithm 1 in Appendix E.

### 3.1. Incremental Graph Learning for Large Datasets

Real-world graph data is continuously expanding. For instance, e-commerce networks accumulate new clicks and purchases daily (Xiang et al., 2010), while academic networks grow with new researchers and publications (Wang et al., 2020). This expanding behaviour suggests that large graphs can be efficiently processed by learning them incrementally in smaller segments.

Given a large dataset $\mathcal{L}(V_\mathcal{L}, X_\mathcal{L})$, where $V_\mathcal{L}$ is the node set and $X_\mathcal{L}$ represents node features, we define an *expanding dataset* setting $\mathcal{L}_\mathcal{E} = \{\mathcal{E}_{\tau=0}^T\}$. Initially, $\mathcal{L}$ is split into: (i) a *static dataset* $\mathcal{E}_0(V_0, X_0)$ and (ii) an *expanding dataset* $\mathcal{E} = \{\mathcal{E}_\tau(V_\tau, X_\tau)\}_{\tau=1}^T$. Both *Goal 1* (large datasets with missing graph structure) and *Goal 2* (partially available graphs with incremental updates), discussed in Section 2, share the common objective of incrementally learning and updating the graph structure as new data arrives. Graph-FLEx handles these by decomposing the problem into two key components:

- **Initial Graph** $\mathcal{G}_0(V_0, A_0, X_0)$**:** For *Goal 1*, where the graph structure is entirely missing, $\mathcal{E}_0(V_0, X_0)$ is used to construct $\mathcal{G}_0$ from scratch using structure learning methods (see Section 2.2). For *Goal 2*, the initial graph $\mathcal{G}_0(V_0, A_0, X_0)$ is already available and serves as the starting point for incremental updates.
- **Expanding Dataset** $\mathcal{E} = \{\mathcal{E}_\tau(V_\tau, X_\tau)\}_{\tau=1}^T$**:** In both cases, $\mathcal{E}$ consists of incoming nodes and features arriving over $T$ timestamps. These nodes are progressively integrated into the existing graph, enabling continuous adaptation and growth.

The partition is controlled by a parameter $r$, which determines the proportion of static nodes: $r = \frac{\|V_0\|}{\|V_\mathcal{L}\|}$. For example, $r = 0.2$ implies that 20% of $V_\mathcal{L}$ is treated as static, while the remaining 80% arrives incrementally over $T$ timestamps. In our experiments, we set $r = 0.5$ and $T = 25$.

*Remark* 2. We can learn $\mathcal{G}_\tau(V_\tau, A_\tau, X_\tau)$ by aggregating $\mathcal{E}_\tau$ nodes in $\mathcal{G}_{\tau-1}$ graph. Our goal is to learn $\mathcal{G}_T(V_T, A_T, X_T)$ after $T^{th}$-timestamp.

### 3.2. Detecting Communities

From the static graph $\mathcal{G}_0$, our goal is to learn higher-order structural information, identifying potential communities to which incoming nodes ($V \in V\tau$) may belong. We train the community detection/clustering model $\mathcal{M}_{\text{clust}}$ once using $\mathcal{G}_0$, allowing subsequent inference of clusters for all incoming nodes. While our framework supports spectral and k-means clustering, our primary focus has been on

Graph Neural Network (GNN)-based clustering methods. Specifically, we use DMoN (Tsitsulin et al., 2023; Bianchi et al., 2020; Bianchi, 2022), which maximizes spectral modularity. Modularity (Newman, 2006) measures the divergence between intra-cluster edges and the expected number. These methods use a GNN layer to compute the partition matrix $C = \text{softmax}(\text{MLP}(\widetilde{X}, \theta_{\text{MLP}})) \in \mathbb{R}^{N \times K}$, where $K$ is the number of clusters and $\widetilde{X}$ is the updated feature embedding generated by one or more message-passing layers. To optimize the $C$ matrix, we minimize the loss function $\Delta(C; A) = -\frac{1}{2m}\text{Tr}(C^T B C) + \frac{\sqrt{k}}{n}|\Sigma_i C_i^T|_F - 1$, which combines spectral modularity maximization with regularization to prevent trivial solutions, where $B$ is the modularity matrix (Tsitsulin et al., 2023). Our static graph $\mathcal{G}_0$ and incoming nodes $\mathcal{E}$ follow Assumption 1.

**Assumption 1.** *We assume that the generated graphs adhere to the Degree-Corrected Stochastic Block Model (DC-SBM) (Zhao et al., 2012), where intra-class (or intra-community) links are more likely than inter-class links.*

For more details on DC-SBM, see Appendix A.

**Lemma 1.** $\mathcal{M}_{clust}$ *Consistency. We adopt the theoretical framework of (Zhao et al., 2012) for a DC-SBM with $N$ nodes and $k$ classes. The edge probability matrix is parameterized as $P_N = \rho_N P$, where $P \in \mathbb{R}^{k \times k}$ is a symmetric matrix containing the between/within community edge probabilities and it is independent of $N$, $\rho_N = \lambda_N/N$, and $\lambda_N$ is the average degree of the network. Let $\hat{y}_N = [\hat{y}_1, \hat{y}_2, \ldots, \hat{y}_N]$ denote the predicted class labels, and let $\hat{C}_N$ be the corresponding $N \times k$ one-hot matrix. Let the true class label matrix is $C_N$, and $\mu$ is any $k \times k$ permutation matrix. Under the adjacency matrix $A^{(N)}$, the global maximum of the objective $\Delta(\cdot; A^{(N)})$ is denoted as $\hat{C}_N^*$. The consistency of class predictions is defined as:*

1. *Strong Consistency.*
$$P_N\left[\min_\mu \|\hat{C}_N^*\mu - C_N\|_F^2 = 0\right] \to 1 \quad as\ N \to \infty,$$

2. *Weak Consistency.*
$$\forall \varepsilon > 0, P_N\left[\min_\mu \frac{1}{N}\|\hat{C}_N^*\mu - C_N\|_F^2 < \varepsilon\right] \to 1\ as\ N \to \infty.$$

*where $\|\cdot\|_F$ is the Frobenius norm. Under the conditions of Theorem 3.1 from (Zhao et al., 2012):*

- *The $\mathcal{M}_{clust}$ objective is strongly consistent if $\lambda_N/\log(N) \to \infty$, and*
- *It is weakly consistent when $\lambda_N \to \infty$.*

*Remark* 3. **Structure Learning within Communities.** In *GraphFLEx*, we focus on learning the structure within each community rather than the structure of the entire dataset at once. Strong consistency ensures perfect community recovery, meaning no inter-community edges exist

representing the ideal case. Weak consistency, however, allows for a small fraction ($\epsilon$) of inter-community edges, where $\epsilon$ is controlled by $\rho_n$ in $P_n = \rho_n P$, influencing graph sparsity.

By Lemma 1 and Assumption 1, stronger consistency leads to more precise structure learning, whereas weaker consistency permits a limited number of inter-community edges.

### 3.3. Learning Graph Structure on a Coarse Graph

After training $\mathcal{M}_{\text{clust}}$, we identify communities for incoming nodes, starting with $\tau = 1$. Once assigned, we determine significant communities those with at least one incoming node and learn their connections to the respective community subgraphs. For large datasets, substantial community sizes may again introduce scalability issues. To mitigate this, we first coarsen the large community graph into a smaller graph and use it to identify potential connections for incoming nodes. This process constitutes the second module of GraphFLEx, denoted as $\mathcal{M}_{\text{coar}}$, which employs LSH-based hashing for graph coarsening. The supernode index for $i^{th}$ node is given as:

$$\mathcal{H}_i = maxOccurance \left\{ \left\lfloor \frac{1}{r} \cdot (\mathcal{W} \cdot X_i + b) \right\rfloor \right\} \quad (1)$$

where $r$ (bin width) controls the coarsened graph size, $\mathcal{W}$ represents random projection matrix, $X$ is the feature matrix, and $b$ is the bias term. For further details, refer to UGC (Kataria et al., 2024). After coarsening the $i^{th}$ community ($C_i$), $\mathcal{M}_{\text{coar}}(C_i) = \{\mathcal{P}_i, S_i\}$ yields a partition matrix $\mathcal{P}_i \in \mathbb{R}^{\|S_i\| \times \|C_i\|}$ and a set of coarsened supernodes ($S_i$), as discussed in Section 2.

**Definition 1.** *The neighborhood of a set of nodes $\mathcal{E}_i$ is defined as the union of the top $k$ most similar nodes in $C_i$ for each node $v \in \mathcal{E}_i$, where similarity is measured by the distance function $d(v, u)$. A node $u \in C_i$ is considered part of the neighborhood if its distance $d(v, u)$ is among the $k$ smallest distances for all $u' \in C_i$.*

$$\mathcal{N}_k(\mathcal{E}_i) = \bigcup_{v \in \mathcal{E}_i} \{u \in C_i \mid d(v, u) \leq top\text{-}k[d(v, u') : u' \in C_i]\}$$

**Goal 3.** *The neighborhood of incoming nodes $\mathcal{N}_k(\mathcal{E}_i)$ represents the ideal set of nodes where the incoming nodes $\mathcal{E}_i$ are likely to establish connections when the entire community is provided to a structure learning framework.. A robust coarsening framework must reduce the number of nodes within each community $C_i$ while ensuring that the neighborhood of the incoming nodes is preserved.*

### 3.4. Graph Learning only with Potential Nodes

As we now have a smaller representation of the community, we can employ any graph learning algorithms discussed in

Section 2.2 to learn a graph between coarsened supernodes $S_i$ and incoming nodes ($V_\tau^i \in V_\tau$). This is the third module of GraphFLEx, i.e., graph learning; we denote it as $\mathcal{M}_{gl}$. The number of supernodes in $S_i$ is much smaller compared to the original size of the community, i.e., $\|S_i\| \ll \|C_i\|$; scalability is not an issue now. We learn a small graph first using $\mathcal{M}_{gl}(S_i, X_\tau^i) = \widetilde{\mathcal{G}}_\tau^i(V_\tau^c, A_\tau^c)$ where $X_\tau^i$ represents features of new nodes belonging to $i^{th}$ community at time $\tau$, $\widetilde{\mathcal{G}}_\tau^i(V_\tau^c, A_\tau^c)$ representing the graph between supernodes and incoming nodes. Utilizing the partition matrix $\mathcal{P}_i$ obtained from $\mathcal{M}_{\text{coar}}$, we can precisely determine the set of nodes associated with each supernode. For every new node $V \in V_\tau^i$, we identify the connected supernodes and subsequently select nodes within those supernodes. This subset of nodes is denoted by $\omega_{V_\tau^i}$, the sub-graph associated with $\omega_{V_\tau^i}$ represented by $\mathcal{G}_{\tau-1}^i(\omega_{V_\tau^i})$ then undergoes an additional round of graph learning $\mathcal{M}_{gl}(\mathcal{G}_{\tau-1}^i(\omega_{V_\tau^i}), X_\tau^i)$, ultimately providing a clear and accurate connection of new nodes $V_\tau^i$ with nodes of $\mathcal{G}_{\tau-1}$, ultimately updating it to $\mathcal{G}_\tau$. This multi-step approach, characterized by coarsening, learning on coarsened graphs, and translation to the original graph, ensures scalability.

**Theorem 1.** *Neighborhood Preservation. Let $\mathcal{N}_k(\mathcal{E}_i)$ denote the neighborhood of incoming nodes $\mathcal{E}_i$ for the $i^{th}$ community. With partition matrix $\mathcal{P}_i$ and $\mathcal{M}_{gl}(S_i, X_\tau^i) = \mathcal{G}_\tau^c(V_\tau^c, A_\tau^c)$ we identify the supernodes connected to incoming nodes $\mathcal{E}_i$ and subsequently select nodes within those supernodes; this subset of nodes is denoted by $\omega_{V_\tau^i}$. Formally,*

$$\omega_{V_\tau^i} = \bigcup_{v \in \mathcal{E}_i} \left\{ \bigcup_{s \in S_i} \{\pi^{-1}(s) | A_\tau^c(v, s) \neq 0\} \right\}$$

*Then, with probability $\Pi_{\{c \in \phi\}} p(c)$, it holds that $\mathcal{N}_k(\mathcal{E}_i) \subseteq \omega_{V_\tau^i}$ where*

$$p(c) \leq 1 - \frac{2}{\sqrt{2\pi}} \frac{c}{r} \left[ 1 - e^{-r^2/(2c^2)} \right],$$

*and $\phi$ is a set containing all pairwise distance values ($c = \|v - u\|$) between every node $v \in \mathcal{E}_i$ and the nodes $u \in \omega_{V_\tau^i}$. Here, $\pi^{-1}(s)$ denotes the set of nodes mapped to supernode $s$, $r$ is the bin-width hyperparameter of $\mathcal{M}_{\text{coar}}$.*

*Proof.* The proof is deferred in Appendix B. □

*Remark* 4. Theorem 1 establishes that, with a constant probability of success, the neighborhood of incoming nodes $\mathcal{N}_k(\mathcal{E}_i)$ can be effectively recovered using the GraphFLEx multistep approach, which involves coarsening and learning on the coarsened graph, i.e., $\mathcal{N}_k(\mathcal{E}_i) \subseteq \omega_{V_\tau^i}$. The set $\omega_{V_\tau^i}$, estimated by GraphFLEx, identifies potential candidates where incoming nodes are likely to connect. The probability of failure can be reduced by regulating the average degree of connectivity in $\mathcal{M}_{gl}(S_i, X_\tau^i) = \mathcal{G}_\tau^c(V_\tau^c, A_\tau^c)$. While a fully connected $\mathcal{G}_\tau^c$ ensures all nodes in the community are candidates, it significantly increases computational costs for large communities.

Table 2: Time complexity analysis of GraphFLEx. Here, $N$ is the number of nodes in the graph, $k$ is the number of nodes in the static subgraph used for clustering ($k \ll N$), and $c$ represents the number of detected communities. $k_\tau$ denotes the number of nodes at timestamp $\tau$. Finally, $\alpha = \|S_\tau^i\| + \|\mathcal{E}_\tau^i\|$ is the sum of coarsened and incoming nodes in the relevant community at $\tau$ timestamp.

|  | $\mathcal{M}_{clust}$ | $\mathcal{M}_{coar}$ | $\mathcal{M}_{gl}$ | GraphFLEx |
|---|---|---|---|---|
| **Best (kNN-UGC-ANN)** | $\mathcal{O}(k^2)$ | $\mathcal{O}\left(\frac{k_\tau}{c}\right)$ | $\mathcal{O}(\alpha \log \alpha)$ | $\mathcal{O}(k^2 + \frac{k_\tau}{c} + \alpha \log \alpha)$ |
| **Worst (SC-FGC-GLasso)** | $\mathcal{O}(k^3)$ | $\mathcal{O}\left(\left(\frac{k_\tau}{c}\right)^2 \|S_\tau^i\|\right)$ | $\mathcal{O}(\alpha^3)$ | $\mathcal{O}(k^3 + \left(\frac{k_\tau}{c}\right)^2 \|S_\tau^i\| + \alpha^3)$ |

### 3.5. GraphFLEx Offering Multiple SGL Frameworks

Each module in Figure 3, controls distinct properties: clustering influences community detection, coarsening governs supernode formation to reduce graph complexity, and the learning module enforces diverse structural properties. Altering any of these modules results in a new graph learning method. Currently, we support 48 different graph learning configurations, and this number scales exponentially with the addition of new methods to any module. The number of possible frameworks is given by $\alpha \times \beta \times \gamma$, where $\alpha$, $\beta$, and $\gamma$ represent the number of clustering, coarsening, and learning methods, respectively.

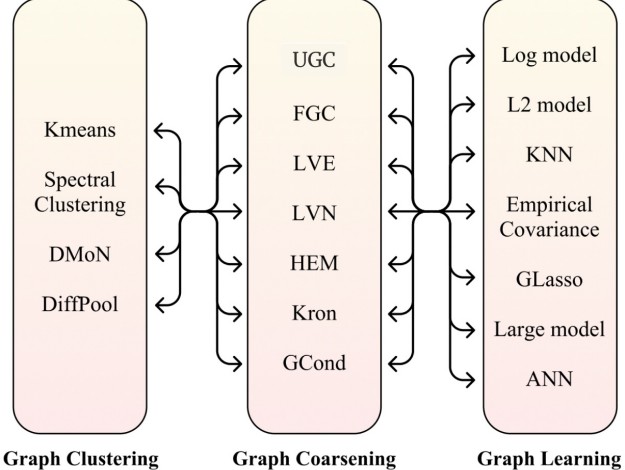

**Graph Clustering**    **Graph Coarsening**    **Graph Learning**

Figure 3: The versatility of GraphFlex in supporting multiple methods for structure learning.

### 3.6. Run Time Analysis

We evaluate the run-time complexity of GraphFLEx in two scenarios: (a) the worst-case scenario, where computationally intensive clustering and coarsening modules are selected, providing an upper bound on time complexity, and (b) the best-case scenario, where the most efficient modules are chosen. Table 2 summarizes the analysis. The run time of GraphFLEx is primarily determined by the learning module ($\mathcal{M}_{gl}$). GraphFLEx computational time is always bounded by existing approaches, as it operates on a significantly reduced graph space, ensuring efficient performance, especially for larger or expanding graphs. This is also illustrated in Table 3.

## 4. Experiments

In this section, we conclude the experiments to back up our findings.

**Tasks and Datasets.** The experiments focus on four key aspects of GraphFLEx: its computational efficiency, scalability in handling large graphs, the quality of the learned graph structure, and its ability to efficiently handle expanding graphs. To validate the characteristics of GraphFLEx, we conduct extensive experiments on 22 different datasets, including (a) datasets that already have a complete graph structure (allowing comparison between the learned and the original structure), (b) datasets with missing graph structures, (c) synthetic datasets, and (d) small datasets for visualizing the graph structure. More details about datasets are presented in Table 6 in Appendix D.

*System Specifications:* All the experiments conducted for this work were performed on an Intel Xeon W-295 CPU and 64GB of RAM desktop using the Python environment.

**Computational Efficiency.** Existing methods like $k$-NN and $log$-model struggle to learn graph structures even for 20k nodes due to out-of-memory (OOM) or out-of-time (OOT) issues, while $l2$-model and $large$-model struggle beyond 50k nodes. Although $A$-NN and $emp$-Covar. are faster, GraphFLEx outperforms them on sufficiently large graphs (Table 3). While traditional methods may be efficient for small graphs, GraphFLEx scales significantly better, excelling on large datasets like *Pubmed* and *Syn 5*, where most methods fail. It accelerates structure learning, making $A$-NN 3× faster and $emp$-Covar. 2× faster.

### 4.1. Node Classification Accuracy

**Experimental Setup.** We now evaluate the prediction performance of GNN models when trained on graph structures learned from three distinct scenarios: **1) Original Structure:** GNN models trained on the original graph structure, which we refer to as the Base Structure, **2) GraphFLEx Structure:** GNN models trained on the graph structure learned from GraphFLEx, and **3)Vanilla Structure:** GNN models trained on the graph structure learned from other existing methods.

For each scenario, a unique graph structure is obtained. We trained GNN models on each of these three structure. For more details on GNN model parameters, see Appendix F.

Table 3: Computational time for learning graph structures using GraphFLEx (GFlex) with existing methods (Vanilla referred to as Van.). The experimental setup involves treating 50% of the data as static, while the remaining 50% of nodes are treated as incoming nodes arriving in 25 different timestamps. The best times are highlighted by color Green. OOM and OOT denote out-of-memory and out-of-time, respectively.

| Data | ANN | | KNN | | log-model | | l2-model | | emp-Covar. | | large-model | |
|------|------|-------|------|-------|-----------|-------|----------|-------|------------|-------|-------------|-------|
| | Van. | GFlex | Van. | GFlex | Van. | GFlex | Van. | GFlex | Van. | GFlex | Van. | GFlex |
| Cora | 335 | 100 | 8.4 | 36.1 | 869 | 81.6 | 424 | 55 | 8.6 | 30 | 2115 | 18.4 |
| Citeseer | 1535 | 454 | 21.9 | 75 | 1113 | 64.5 | 977 | 54.0 | 14.7 | 59.2 | 8319 | 43.9 |
| DBLP | 2731 | 988 | OOM | 270 | 77000 | 919 | OOT | 1470 | 359 | 343 | OOT | 299 |
| CS | 22000 | 12000 | OOM | 789 | OOT | 838 | 32000 | 809 | 813 | 718 | OOT | 1469 |
| PubMed | 770 | 227 | OOM | 164 | OOT | 176 | OOT | 165 | 488 | 299 | OOT | 262 |
| Phy. | 61000 | 21000 | OOM | 903 | OOT | 959 | OOT | 908 | 2152 | 1182 | OOT | 2414 |
| Syn 3 | 95 | 37 | OOM | 30 | 58000 | 346 | 859 | 53 | 88 | 59 | 5416 | 42 |
| Syn 4 | 482 | 71 | OOM | 73 | OOT | 555 | OOT | 145 | 2072 | 1043 | OOT | 392 |

Table 4: Node classification accuracies on different GNN models using GraphFLEx (GFlex) with existing Vanilla (Van.) methods. The experimental setup involves treating 70% of the data as static, while the remaining 30% of nodes are treated as new nodes coming in 25 different timestamps. The best and the second-best accuracies in each row are highlighted by dark and lighter shades of Green, respectively. GraphFLEx's structure beats all of the vanilla structures for every dataset. OOM and OOT denotes out-of-memory and out-of-time respectively.

| Data | Model | ANN | | KNN | | log-model | | l2-model | | COVA | | large-model | | Base Struct. |
|------|-------|------|-------|------|-------|-----------|-------|----------|-------|------|-------|-------------|-------|--------------|
| | | Van. | GFlex | Van. | GFlex | Van. | GFlex | Van. | GFlex | Van. | GFlex | Van. | GFlex | |
| DBLP | GAT | 34.23 | 67.37 | OOM | 69.83 | OOT | 69.83 | OOT | 68.98 | 50.48 | 68.56 | OOT | 66.38 | 70.84 |
| | SAGE | 34.23 | 69.58 | OOM | 70.28 | OOT | 70.28 | OOT | 70.68 | 51.47 | 70.51 | OOT | 69.32 | 72.57 |
| | GCN | 34.12 | 69.41 | OOM | 73.39 | OOT | 73.39 | OOT | 73.05 | 51.50 | 71.75 | OOT | 68.55 | 74.43 |
| | GIN | 34.01 | 69.69 | OOM | 68.19 | OOT | 68.19 | OOT | 73.08 | 52.77 | 72.03 | OOT | 71.18 | 73.92 |
| CS | GAT | 12.47 | 60.89 | OOM | 61.09 | OOT | 60.95 | 18.64 | 61.06 | 58.96 | 88.06 | OOT | 86.22 | 60.75 |
| | SAGE | 12.70 | 78.81 | OOM | 79.43 | OOT | 79.06 | 19.24 | 78.94 | 56.97 | 93.30 | OOT | 92.79 | 80.33 |
| | GCN | 12.59 | 63.81 | OOM | 67.94 | OOT | 69.33 | 19.21 | 66.01 | 58.35 | 91.07 | OOT | 84.85 | 67.43 |
| | GIN | 13.07 | 77.62 | OOM | 78.41 | OOT | 78.55 | 19.24 | 77.61 | 58.26 | 92.07 | OOT | 86.03 | 55.65 |
| Pub. | GAT | 49.49 | 83.71 | OOM | 84.60 | OOT | 84.60 | OOT | 84.04 | 72.63 | 83.97 | OOT | 81.15 | 84.04 |
| | SAGE | 50.43 | 87.27 | OOM | 87.34 | OOT | 87.34 | OOT | 87.42 | 73.57 | 86.68 | OOT | 87.34 | 88.88 |
| | GCN | 50.45 | 82.06 | OOM | 83.56 | OOT | 83.56 | OOT | 83.74 | 73.14 | 82.39 | OOT | 78.03 | 85.54 |
| | GIN | 51.82 | 83.13 | OOM | 84.31 | OOT | 84.07 | OOT | 82.93 | 73.15 | 83.51 | OOT | 82.85 | 86.50 |
| Phy. | GAT | 29.18 | 88.06 | OOM | 88.47 | OOT | 88.47 | OOT | 88.68 | 58.96 | 88.06 | OOT | 86.22 | 88.58 |
| | SAGE | 29.57 | 93.47 | OOM | 93.47 | OOT | 93.47 | OOT | 93.78 | 56.97 | 93.60 | OOT | 92.79 | 94.19 |
| | GCN | 27.84 | 91.27 | OOM | 91.08 | OOT | 91.08 | OOT | 91.78 | 58.35 | 91.07 | OOT | 84.85 | 91.48 |
| | GIN | 28.38 | 92.69 | OOM | 92.04 | OOT | 92.04 | OOT | 92.27 | 58.26 | 92.07 | OOT | 86.03 | 88.89 |

**GNN Models.** Graph neural networks (GNNs) such as $GCN$ (Kipf & Welling, 2016), $GraphSage$ (Hamilton et al., 2017), $GIN$ (Xu et al., 2018), and $GAT$ (Velickovic et al., 2017) rely on accurate message passing, dictated by the graph structure, for effective embedding. We use these models to evaluate the above-mentioned learned structures. Table 4 reports node classification performance across all methods. Notably, GraphFLEx outperforms vanilla structures by a significant margin across all datasets, achieving accuracies close to those obtained with the original structure. Figure 9 in Appendix F illustrates $GraphSage$ classification results, highlighting GraphFLEx's superior performance. For the $CS$ dataset, GraphFLEx ($large$-model) and GraphFLEx ($empCovar.$-model) even surpass the original

structure, demonstrating its ability to preserve key structural properties while denoising edges, leading to improved accuracy.

### 4.2. Clustering Quality

We measure three metrics to evaluate the resulting clusters or community assignments: a) Normalized Mutual Information (NMI) (Tsitsulin et al., 2023) between the cluster assignments and original labels; b) Conductance ($\mathcal{C}$) (Jerrum & Sinclair, 1988) which measures the fraction of total edge volume that points outside the cluster; and c) Modularity ($\mathcal{Q}$) (Newman, 2006) which measures the divergence between the intra-community edges and the expected one. Table 5

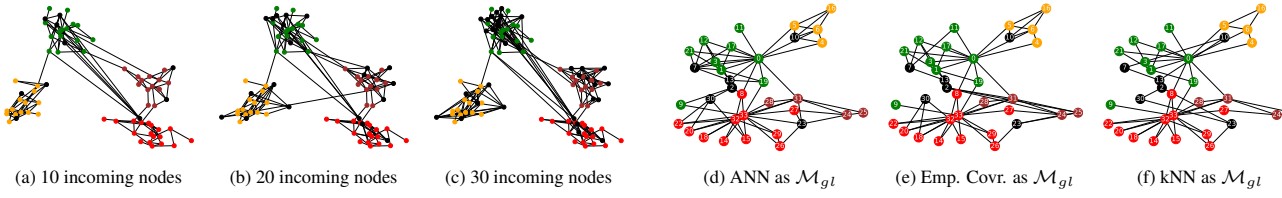

| (a) 10 incoming nodes | (b) 20 incoming nodes | (c) 30 incoming nodes | (d) ANN as $\mathcal{M}_{gl}$ | (e) Emp. Covr. as $\mathcal{M}_{gl}$ | (f) kNN as $\mathcal{M}_{gl}$ |

Figure 4: Figures (a), (b), and (c) illustrate the growing structure learned using GraphFLEx for *HE* synthetic dataset. Figures (d), (e), and (f) illustrate the learned structure on Zachary's karate dataset when existing methods are employed with GraphFLEx. New nodes are denoted using black color.

illustrates these metrics for single-cell RNA and MNIST dataset (where the whole structure is missing), and Figure 5 shows the PHATE (Moon et al., 2019) visualization of clusters learned using GraphFLEx's clustering module $\mathcal{M}_{clust}$. We also train the aforementioned GNN models for the node classification task in order to illustrate the efficacy of the learned structures; the accuracies values presented in Table 5, clearly highlight the significance of the learned structures, as reflected by the high accuracy values.

Table 5: Clustering results and node classification accuracies. Left: Clustering metrics - NMI, graph conductance $C$, and Modularity $\mathcal{Q}$. Right: Node classification accuracy for GCN, GraphSAGE, GIN, GAT.

| **Data** | $NMI\uparrow$ | $\mathcal{C}\downarrow$ | $\mathcal{Q}\uparrow$ | **GCN** | **SAGE** | **GIN** | **GAT** |
|---|---|---|---|---|---|---|---|
| Bar. M. | 0.716 | 0.057 | 0.741 | 91.2 | 96.2 | 95.1 | 94.9 |
| Seger. | 0.678 | 0.102 | 0.694 | 91.0 | 93.9 | 94.2 | 92.3 |
| Mura. | 0.843 | 0.046 | 0.706 | 96.9 | 97.4 | 97.5 | 96.4 |
| Bar. H. | 0.674 | 0.078 | 0.749 | 95.3 | 96.4 | 97.2 | 95.8 |
| Xin | 0.741 | 0.045 | 0.544 | 98.6 | 99.3 | 98.9 | 99.8 |
| MNIST | 0.677 | 0.082 | 0.712 | 92.9 | 94.5 | 94.9 | 82.6 |

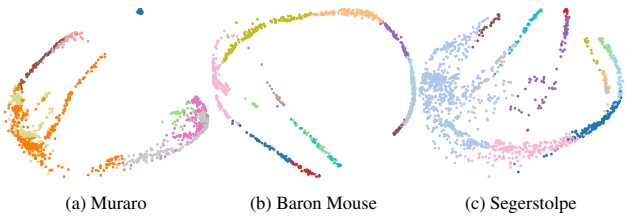

| (a) Muraro | (b) Baron Mouse | (c) Segerstolpe |

Figure 5: PHATE visualization of clusters learned using Graph-FLEx clustering module for scRNA-seq datasets.

### 4.3. Structure Visualization

We evaluate the structures generated by GraphFLEx through visualizations on four small datasets: (i) MNIST (LeCun et al., 2010), consisting of handwritten digit images, where Figure 6(a) shows that images of the same digit are mostly connected; (ii) Pre-trained GloVe embeddings (Pennington et al., 2014) of English words, with Figure 6(b) revealing that frequently used words are closely connected; (iii) A synthetic *H.E* dataset (see Appendix D), demonstrating Graph-FLEx's ability to handle expanding networks without requir-

ing full relearning. Figure 4(a-c) shows the graph structure evolving as 30 new nodes are added over three timestamps; and (iv) Zachary's karate club network (Zachary, 1977), which highlights GraphFLEx's multi-framework capability. Figure 4(d-f) shows three distinct graph structures after altering the learning module.

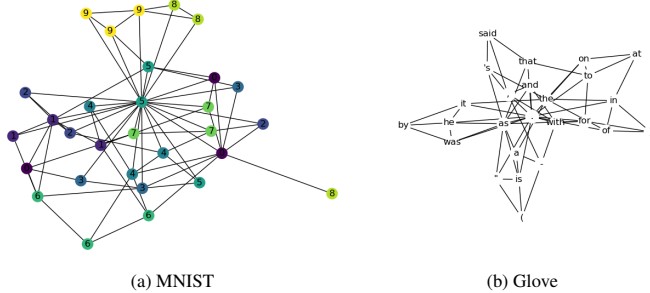

| (a) MNIST | (b) Glove |

Figure 6: Figures demonstrate the effectiveness of our framework in learning meaningful structure between similar MNIST digit images and pre-trained GloVe embeddings.

## 5. Conclusion

Large or expanding graphs challenge the best of graph learning approaches. GraphFLEx, introduced in this paper, seamlessly adds new nodes into an existing graph structure. It offers diverse methods for acquiring the graph's structure. GraphFLEx consists of three key modules: Clustering, Coarsening, and Learning which empowers Graph-FLEx to serves as a comprehensive framework applicable individually for clustering, coarsening, and learning tasks. GraphFLEx is typically 3X faster than other state of the art methods and scales well with large graphs. It achieves accuracies close to training on the original graph, in most instances. The performance across multiple real and synthetic datasets affirms the utility and efficacy of GraphFLEx for graph structure learning.

**Limitations and Future Work.** GraphFLEx is designed assuming minimal inter-community connectivity, which aligns well with many real-world scenarios. However, its applicability to heterophilic graphs may require further adaptation. Future work will focus on extending the framework to supervised GSL methods and heterophilic graphs, broadening its scalability and versatility.

## Impact Statement

This paper presents work whose goal is to advance the field of Machine Learning. There are many potential societal consequences of our work, none which we feel must be specifically highlighted here.

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

# Appendix

## A. Degree-Corrected Stochastic Block Model(DC-SBM)

The DC-SBM is one of the most commonly used models for networks with communities and postulates that, given node labels $\mathbf{c} = c_1, ... c_n$, the edge variables $A'_{ij}s$ are generated via the formula

$$E[A_{ij}] = \theta_i \theta_j P_{c_i} P_{c_j}$$

, where $\theta_i$ is a "degree parameter" associated with node $i$, reflecting its individual propernsity to form ties, and $P$ is a $K \times K$ symmetric matrix containing the between/withincommunity edge probabilities and $P_{c_i} P_{c_j}$ denotes the edge probabilities between community $c_i$ and $c_j$.

For DC-SBM model (Zhao et al., 2012) assumed $P_n$ on $n$ nodes with $k$ classes, each node $v_i$ is given a label/degree pair$(c_i, \theta_i)$, drawn from a discrete joint distribution $\Pi_{K \times m}$ which is fixed and does not depend on n. This implies that each $\theta_i$ is one of a fixed set of values $0 \leq x_1 \leq .... \leq x_m$. To facilitate analysis of asymptotic graph sparsity, we parameterize the edge probability matrix $P$ as $P_n = \rho_n P$ where P is independent of $n$, and $\rho_n = \lambda_n / n$ where $\lambda_n$ is the average degree of the network.

## B. Neighbourhood Preservation

**Theorem 2.** *Neighborhood Preservation. Let $\mathcal{N}_k(\mathcal{E}_i)$ denote the neighborhood of incoming nodes $\mathcal{E}_i$ for the $i^{th}$ community. With partition matrix $\mathcal{P}_i$ and $\mathcal{M}_{gl}(S_i, X^i_\tau) = \mathcal{G}^c_\tau(V^c_\tau, A^c_\tau)$ we identify the supernodes connected to incoming nodes $\mathcal{E}_i$ and subsequently select nodes within those supernodes; this subset of nodes is denoted by $\omega_{V^i_\tau}$. Formally,*

$$\omega_{V^i_\tau} = \bigcup_{v \in \mathcal{E}_i} \left\{ \bigcup_{s \in S_i} \{\pi^{-1}(s) | A^c_\tau(v, s) \neq 0\} \right\}$$

*Then, with probability $\Pi_{\{c \in \phi\}} p(c)$, it holds that $\mathcal{N}_k(\mathcal{E}_i) \subseteq \omega_{V^i_\tau}$ where*

$$p(c) \leq 1 - \frac{2}{\sqrt{2\pi}} \frac{c}{r} \left[ 1 - e^{-r^2/(2c^2)} \right],$$

*and $\phi$ is a set containing all pairwise distance values ($c = \|v - u\|$) between every node $v \in \mathcal{E}_i$ and the nodes $u \in \omega_{V^i_\tau}$. Here, $\pi^{-1}(s)$ denotes the set of nodes mapped to supernode $s$, $r$ is the bin-width hyperparameter of $\mathcal{M}_{coar}$.*

**Proof:** The probability that LSH random projection (Kataria et al., 2024; Datar et al., 2004) preserves the distance between two nodes $v$ and $u$ i.e., $d(u, v) = c$, is given by:

$$p(c) = \int_0^r \frac{1}{c} f_2 \left( \frac{t}{c} \right) \left( 1 - \frac{t}{r} \right) dt,$$

where $f_2(x) = \frac{2}{\sqrt{2\pi}} e^{-x^2/2}$ represents the Gaussian kernel when the projection matrix is randomly sampled from $p$-stable($p = 2$) distribution (Datar et al., 2004).

The probability $p(c)$ can be decomposed into two terms:

$$p(c) = S_1(c) - S_2(c),$$

$S_1(c)$ and $S_2(c)$ are defined as follows:

$$S_1(c) = \frac{2}{\sqrt{2\pi}} \int_0^r e^{-(t/c)^2/2} dt \leq 1,$$

$$S_2(c) = \frac{2}{\sqrt{2\pi}} \int_0^r e^{-(t/c)^2/2} \frac{t}{r} dt.$$

$$S_2(c) = \frac{2}{\sqrt{2\pi}} \cdot \frac{c}{r} \int_0^r e^{-(t/c)^2/2} \frac{t}{c^2} dt$$

Expanding $S_2(c)$ :

$$S_2(c) = \frac{2}{\sqrt{2\pi}} \cdot \frac{c}{r} \int_0^{r^2/(2c^2)} e^{-y} dy$$

$$S_2(c) = \frac{2}{\sqrt{2\pi}} \cdot \frac{c}{r} \left[ 1 - e^{-r^2/(2c^2)} \right]$$

Thus, the probability $p(c)$ can be bounded as:

$$p(c) \leq 1 - \frac{2}{\sqrt{2\pi}} \frac{c}{r} \left[ 1 - e^{-r^2/(2c^2)} \right].$$

Now, let $\phi$ be the set of all pairwise distances $d(u,v)$, where $v \in \mathcal{E}_i$ and node$\omega_{V_\tau^i}$. The probability that all nodes in $\mathcal{N}_k(\mathcal{E}_i)$ are preserved within $\omega_{V_\tau^i}$, requires that all distances $c \in \phi$ are also preserved. The probability is then given by:

$$\prod_{c \in \phi} p(c).$$

$$\prod_{c \in \phi} p(c) \leq \prod_{c \in \phi} \left( 1 - \frac{2}{\sqrt{2\pi}} \frac{c}{r} \left[ 1 - e^{-r^2/(2c^2)} \right] \right).$$

## C. Continual Learning and Dynamic Graph Learning

In this subsection, we highlight the key distinctions between Graph Structure Learning (GSL) and related fields to justify our specific selection of related works in Section 2.2. GSL is often confused with topics such as Continual Learning (CL) and Dynamic Graph Learning (DGL).

CL (Van de Ven & Tolias, 2019; Zhang et al., 2022; Parisi et al., 2019) addresses the issue of catastrophic forgetting, where a model's performance on previously learned tasks degrades significantly after training on new tasks. In CL, the model has access only to the current task's data and cannot utilize data from prior tasks. Conversely, DGL (Kim et al., 2022; Wu et al., 2023; You et al., 2022) focuses on capturing the evolving structure of graphs and maintaining updated graph representations, with access to all prior information.

While both *CL and DGL* aim to *enhance model adaptability* to dynamic data, GSL is primarily concerned with generating *high-quality graph structures* that can be leveraged for downstream tasks such as node classification (Kipf & Welling, 2016), link prediction (Lü & Zhou, 2011), and graph classification (Vogelstein et al., 2012). Moreover, in CL and DGL, different tasks typically involve distinct data distributions, whereas GSL assumes a consistent data distribution throughout.

## D. Datasets

Datasets used in our experiments vary in size, with nodes ranging from 1k to 60k. Table 6 lists all the datasets we used in our work. We evaluate our proposed framework $GraphFlex$ on real-world datasets *Cora ,Citeseer, Pubmed* (Yang et al., 2016), *CS, Physics* (Shchur et al., 2018), *DBLP* (Fu et al., 2020), all of which include graph structures. These datasets allow us to compare the learned structures with the originals. Additionally, we utilize single-cell RNA pancreas datasets (Yang et al., 2022), including Baron, Muraro, Segerstolpe, and Xin, where the graph structure is missing. The Baron dataset was downloaded from the Gene Expression Omnibus (GEO) (accession no. GSE84133). The Muraro dataset was downloaded from GEO (accession no. GSE85241). The Segerstolpe dataset was accessed from ArrayExpress (accession no. E-MTAB-5061). The Xin dataset was downloaded from GEO (accession no. GSE81608). We simulate the expanding graph scenario by splitting the original dataset across different $\mathcal{T}$ timestamps. We assumed 50% of the nodes were static, with the remaining nodes arriving as incoming nodes at different timestamps.

**Synthetic datasets:** Different data generation techniques validate that our results are generalized to different settings. Please refer to Table 6 for more details about the number of nodes, edges, features, and classes, $Syn$ denotes the type of synthetic datasets. Figure 7 shows graphs generated using different methods. We have employed three different ways to generate synthetic datasets which are mentioned below:

- **PyGSP(PyGsp):** We used synthetic graphs created by PyGSP (Defferrard et al.) library. PyG-G and PyG-S denotes grid and sensor graphs from PyGSP.

- **Watts–Strogatz's small world(SW):** (Watts & Strogatz, 1998) proposed a generation model that produces graphs with small-world properties, including short average path lengths and high clustering.
- **Heterophily(HE):** We propose a method for creating synthetic datasets to explore graph behavior across a heterophily spectrum by manipulating heterophilic factor $\alpha$, and classes. $\alpha$ is determined by dividing the number of edges connecting nodes from different classes by the total number of edges in the graph.

**Visulization Datasets:** To evaluate, the learned graph structure, we have also included three datasets: (i) MNIST (LeCun et al., 2010), consisting of handwritten digit images; (ii) Pre-trained GloVe embeddings (Pennington et al., 2014) of English words; and (iii) Zachary's karate club network (Zachary, 1977).

| Category | Data | Nodes | Edges | Feat. | Class | Type |
|---|---|---|---|---|---|---|
| Original Structure Known | Cora | 2,708 | 5,429 | 1,433 | 7 | Citation network |
| | Citeseer | 3,327 | 9,104 | 3,703 | 6 | Citation network |
| | DBLP | 17,716 | 52.8k | 1,639 | 4 | Research paper |
| | CS | 18,333 | 163.7k | 6,805 | 15 | Co-authorship network |
| | PubMed | 19,717 | 44.3k | 500 | 3 | Citation network |
| | Physics | 34,493 | 247.9k | 8,415 | 5 | Co-authorship network |
| Original Structure Not Known | Xin | 1,449 | NA | 33,889 | 4 | Human Pancreas |
| | Baron Mouse | 1,886 | NA | 14,861 | 13 | Mouse Pancreas |
| | Muraro | 2,122 | NA | 18,915 | 9 | Human Pancreas |
| | Segerstolpe | 2,133 | NA | 22,757 | 13 | Human Pancreas |
| | Baron Human | 8,569 | NA | 17,499 | 14 | Human Pancreas |
| Synthetic | Syn 1 | 2,000 | 8,800 | 150 | 4 | SW |
| | Syn 2 | 5,000 | 22k | 150 | 4 | SW |
| | Syn 3 | 10,000 | 44k | 150 | 7 | SW |
| | Syn 4 | 50,000 | 220k | 150 | 7 | SW |
| | Syn 5 | 400 | 1,520 | 100 | 4 | PyG-G |
| | Syn 6 | 2,500 | 9,800 | 100 | 4 | PyG-S |
| | Syn 7 | 1,000 | 9,990 | 150 | 4 | HE |
| | Syn 8 | 2,000 | 40k | 150 | 4 | HE |
| Visulization Datasets | MNIST | 60,000 | NA | 784 | 10 | Images |
| | Zachary's karate | 34 | 156 | 34 | 4 | Karate club network |
| | Glove | 2,000 | NA | 50 | NA | GloVe embeddings |

Table 6: Summary of the datasets.

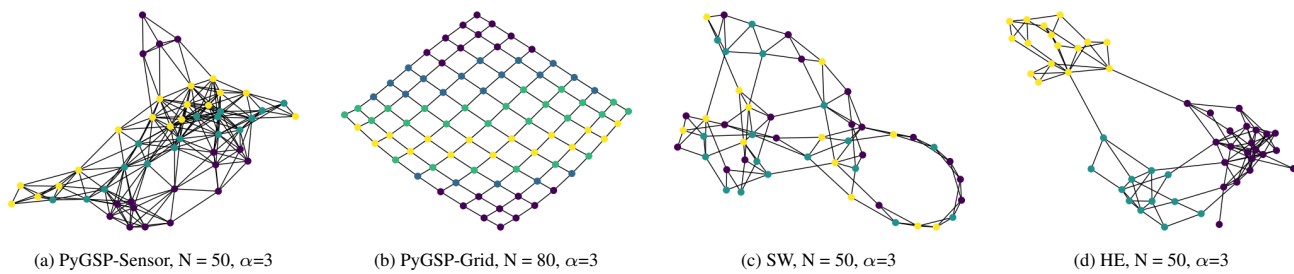

(a) PyGSP-Sensor, N = 50, $\alpha$=3     (b) PyGSP-Grid, N = 80, $\alpha$=3     (c) SW, N = 50, $\alpha$=3     (d) HE, N = 50, $\alpha$=3

Figure 7: This figure illustrates different types of synthetic graphs generated using i)PyGSP, ii) Watts–Strogatz's small world(SW), and iii) Heterophily(HE). N denotes the number of nodes, while $\alpha$ denotes the number of classes.

# E. Algorithm

---

**Algorithm 1** GraphFlex: A Unified Structure Learning framework for expanding and Large Scale Graphs

---

**Input**: Graph $G_0(X_0, A_0)$, expanding nodes set $\mathcal{E}_1^T = \{\mathcal{E}_\tau(\mathcal{V}_\tau, \mathcal{X}_\tau)\}_{\tau=1}^T$
**Parameter**: GClust, GCoar, GL $\leftarrow$ Clustering, Coarsening and Learning Module
**Output**: Graph $G_T(X_T, A_T)$

 1: Train clustering module $train(\mathcal{M}_{clust}, \text{GClust}, G_0)$
 2: **for** each $E_t(V_t, X_t)$ in $\mathcal{E}_1^T$ **do**
 3:    $C_t = infer(\mathcal{M}_{clust}, X_t)$, $C_t \in \mathbb{R}^{N_t}$ denotes the communities of $N_t$ nodes at time $t$.
 4:    $I_t = unique(C_t)$.
 5:    **for** each $I_t^i$ in $I_t$ **do**
 6:       $G_{t-1}^i = \text{subgraph}(G_{t-1}, I_t^i)$
 7:       $\{S_{t-1}^i, P_{t-1}^i\} = \mathcal{M}_{coar}(G_{t-1}^i)$, $S_{t-1}^i \in \mathbb{R}^{k \times d}$ are features of $k$ supernodes, $P_{t-1}^i \in \mathbb{R}^{k \times N_t^i}$ is the partition matrix.
 8:       $Gc_{t-1}^i(S_{t-1}^i, A_{t-1}^i) = \mathcal{M}_{gl}(S_{t-1}^i, X_t^i)$, $Gc_{t-1}^i$ is the learned graph on super-nodes $S_{t-1}^i$ and new node $X_t^i$.
 9:       $\omega_t^i \leftarrow [\,]$
10:       **for** $x \in X_t^i$ **do**
11:          $\omega_t^i.append(x)$
12:          $n_p = \{n \mid A_{t-1}^i[n] > 0\}$
13:          $\omega_t^i.append(n_p)$
14:       **end for**
15:       $G_{t-1} = update(G_{t-1}, \mathcal{M}_{gl}(\omega_t^i))$
16:    **end for**
17:    $G_t = G_{t-1}$
18: **end for**
19: **return** $G_T(X_T, A_T)$

---

## F. Other GNN models

We used four GNN models, namely GCN, GraphSage, GIN, and GAT. Table 7 contains parameter details we used to train GraphFlex. We have used these parameters across all methods.

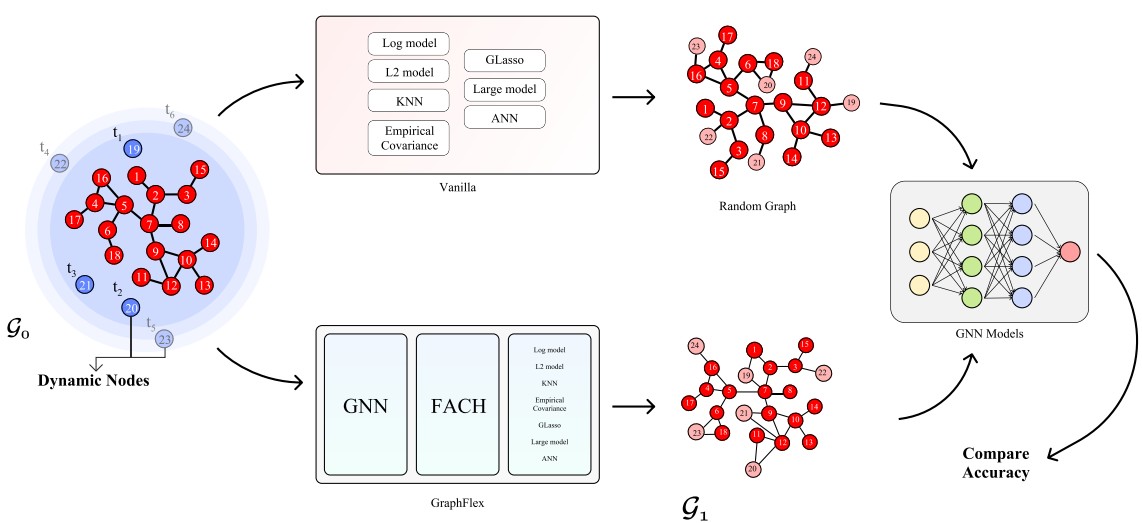

Figure 8: GNN training pipeline.

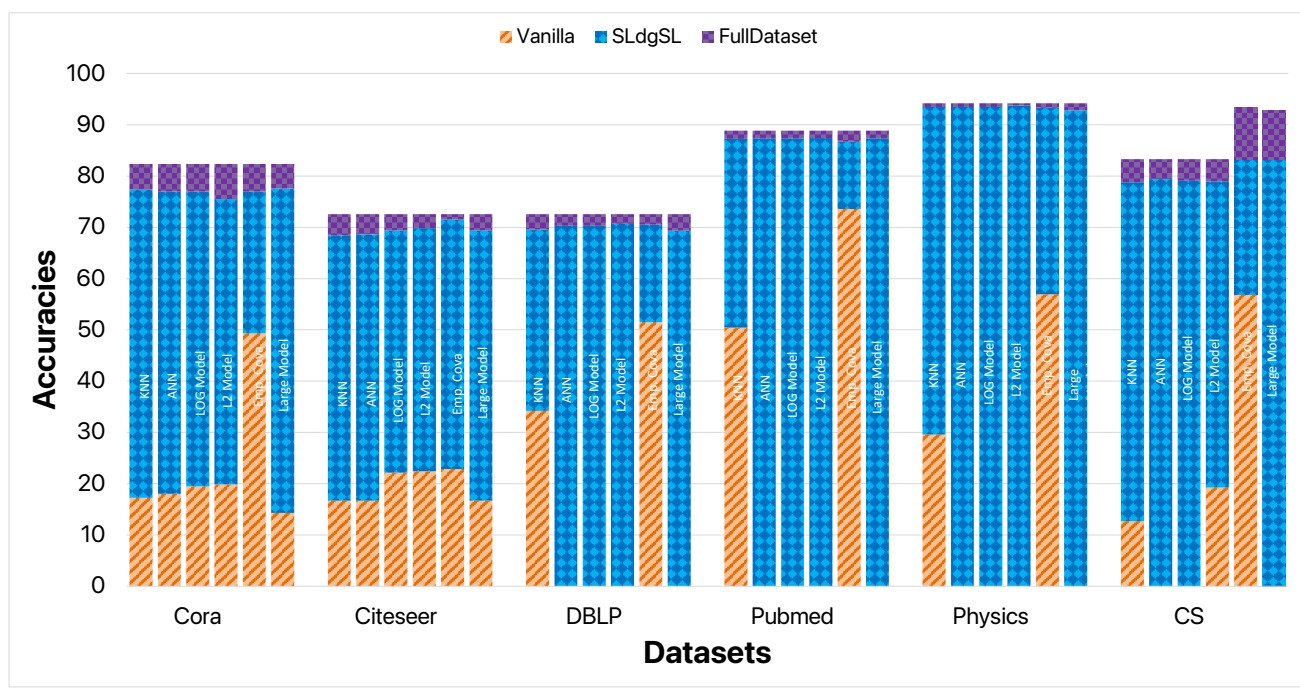

Figure 9: GraphSage accuracies when structure is learned or given with 3 different scenarios(Vanilla, GraphFlex, Original) across different datasets, highlighting performance with 30% node growth over 25 timestamps.

Figure 8 illustrates the pipeline for training our GNN models. Graph structures were learned using both existing methods and GraphFlex, and GNN models were subsequently trained on both structures. Results across all datasets are presented in Table 8 and Table 4.

Table 7: GNN model parameters.

| Model | Hidden Layers | L.R | Decay | Epoch |
|---|---|---|---|---|
| GCN | $\{64, 64\}$ | 0.003 | 0.0005 | 500 |
| GraphSage | $\{64, 64\}$ | 0.003 | 0.0005 | 500 |
| GIN | $\{64, 64\}$ | 0.003 | 0.0005 | 500 |
| GAT | $\{64, 64\}$ | 0.003 | 0.0005 | 500 |

We randomly split data in 60%, 20%, 20% for training-validation-test. The results for these models on synthetic datasets are presented in Table 8.

Figure 8 illustrates the pipeline for training our GNN models. Graph structures were learned using both existing methods and GraphFlex, and GNN models were subsequently trained on both structures.

## G. Computational Efficiency

Table 9 illustrates the remaining computational time for learning graph structures using GraphFLEx with existing Vanilla methods on Synthetic datasets. While traditional methods may be efficient for small graphs, GraphFLEx scales significantly better, excelling on large datasets like *Pubmed* and *Syn 5*, where most methods fail.

Table 8: Node classification accuracies on different GNN models using GraphFLEx (GFlex) with existing Vanilla (Van.) methods. The experimental setup involves treating 70% of the data as static, while the remaining 30% of nodes are treated as new nodes coming in 25 different timestamps. The best and the second-best accuracies in each row are highlighted by dark and lighter shades of Green, respectively. GraphFLEx's structure beats all of the vanilla structures for every dataset. OOM and OOT denotes out-of-memory and out-of-time respectively.

| Dataset | Model | ANN | | KNN | | log-model | | l2-model | | COVA | | large-model | | Base Struc. |
|---|---|---|---|---|---|---|---|---|---|---|---|---|---|---|
| | | Van. | GFlex | Van. | GFlex | Van. | GFlex | Van. | GFlex | Van. | GFlex | Van. | GFlex | |
| Cora | GAT | 18.73 | 73.84 | 20.96 | 73.65 | 16.14 | 72.36 | 18.74 | 73.10 | 49.72 | 77.55 | 14.28 | 76.43 | 79.77 |
| | SAGE | 17.25 | 77.37 | 18.00 | 76.99 | 19.48 | 77.40 | 19.85 | 75.51 | 49.35 | 76.99 | 14.28 | 77.55 | 82.37 |
| | GCN | 17.99 | 78.11 | 17.81 | 77.92 | 18.55 | 77.74 | 20.41 | 79.22 | 47.31 | 80.52 | 14.28 | 79.03 | 84.60 |
| | GIN | 16.69 | 76.44 | 18.74 | 80.52 | 17.44 | 76.25 | 19.29 | 76.62 | 48.79 | 78.85 | 14.28 | 76.06 | 81.63 |
| Citeseer | GAT | 16.51 | 61.82 | 25.00 | 62.27 | 19.24 | 64.70 | 18.18 | 63.48 | 20.91 | 62.73 | 16.67 | 62.27 | 66.42 |
| | SAGE | 16.66 | 68.48 | 16.67 | 68.64 | 22.12 | 69.39 | 22.42 | 69.85 | 22.88 | 71.52 | 16.67 | 69.39 | 72.57 |
| | GCN | 28.18 | 60.00 | 16.67 | 61.97 | 20.45 | 65.45 | 19.70 | 64.24 | 21.06 | 64.70 | 16.67 | 63.18 | 68.03 |
| | GIN | 16.66 | 64.39 | 16.67 | 63.94 | 20.15 | 59.85 | 18.64 | 63.64 | 22.12 | 60.30 | 16.67 | 61.81 | 67.38 |
| Syn 4 | GAT | 29.55 | 92.07 | OOM | 90.86 | OOT | 91.64 | OOT | 91.64 | 35.79 | 92.52 | OOT | 93.74 | 89.49 |
| | SAGE | 26.75 | 87.89 | OOM | 91.05 | OOT | 86.64 | OOT | 86.64 | 32.92 | 90.44 | OOT | 86.01 | 90.03 |
| | GCN | 28.85 | 51.97 | OOM | 19.58 | OOT | 18.29 | OOT | 18.92 | 33.80 | 26.60 | OOT | 36.85 | 21.43 |
| | GIN | 28.50 | 65.61 | OOM | 31.06 | OOT | 26.51 | OOT | 26.56 | 34.03 | 46.40 | OOT | 47.10 | 29.35 |
| Syn 6 | GAT | 44.00 | 86.80 | 43.60 | 86.60 | 30.00 | 78.75 | 55.40 | 92.80 | 36.20 | 93.60 | 31.80 | 92.80 | 97.20 |
| | SAGE | 41.00 | 93.80 | 41.40 | 93.60 | 33.75 | 88.75 | 57.60 | 94.00 | 35.20 | 94.80 | 28.20 | 95.60 | 97.40 |
| | GCN | 43.60 | 88.80 | 42.20 | 87.40 | 26.25 | 81.25 | 55.60 | 92.40 | 31.40 | 94.40 | 25.20 | 94.00 | 99.40 |
| | GIN | 39.60 | 89.00 | 40.40 | 86.60 | 21.25 | 82.50 | 55.20 | 91.80 | 30.00 | 94.60 | 30.40 | 92.00 | 98.80 |
| Syn 8 | GAT | 29.55 | 99.75 | 33.75 | 88.75 | 88.25 | 99.25 | 88.25 | 99.25 | 26.00 | 85.50 | 94.00 | 96.00 | 98.50 |
| | SAGE | 26.75 | 100.0 | 32.50 | 100.0 | 88.75 | 99.50 | 88.75 | 99.50 | 26.75 | 100.0 | 92.50 | 100.0 | 100.0 |
| | GCN | 28.85 | 98.75 | 31.75 | 99.75 | 88.75 | 99.00 | 88.75 | 99.00 | 28.50 | 99.25 | 95.00 | 100.0 | 100.0 |
| | GIN | 28.50 | 50.00 | 30.50 | 91.00 | 82.25 | 91.50 | 82.25 | 91.50 | 27.25 | 81.75 | 91.75 | 92.25 | 78.25 |

Table 9: Computational time for learning graph structures using GraphFLEx (GFlex) with existing methods (Vanilla referred to as Van.). The experimental setup involves treating 50% of the data as static, while the remaining 50% of nodes are treated as incoming nodes arriving in 25 different timestamps. The best times are highlighted by color Green. OOM and OOT denote out-of-memory and out-of-time, respectively.

| Data | ANN | | KNN | | log-model | | l2-model | | COVA | | large-model | |
|---|---|---|---|---|---|---|---|---|---|---|---|---|---|
| | Van. | GFlex | Van. | GFlex | Van. | GFlex | Van. | GFlex | Van. | GFlex | Van. | GFlex |
| Syn 1 | 19.4 | 9.8 | 2.5 | 10.5 | 2418 | 56.4 | 37.2 | 8.8 | 3.5 | 8.3 | 205 | 9.4 |
| Syn 2 | 47.3 | 16.9 | 6.6 | 18.3 | 14000 | 144 | 214 | 22.6 | 20.3 | 18.6 | 1259 | 16.4 |
| Syn 5 | 5.1 | 11.5 | 0.8 | 7.3 | 57.4 | 28 | 1.1 | 5.8 | 0.2 | 4.8 | 3.2 | 5.3 |
| Syn 6 | 16.6 | 9.9 | 2.8 | 11.4 | 1766 | 96.3 | 193 | 101 | 5.3 | 8.9 | 324 | 9.6 |
| Syn 7 | 10.6 | 7.4 | 1.4 | 8.9 | 704 | 85.2 | 10.3 | 7.9 | 0.9 | 6.4 | 36.5 | 8.2 |
| Syn 8 | 19.6 | 11.2 | 2.5 | 11.7 | 2416 | 457 | 37.2 | 17.0 | 3.4 | 10.9 | 204 | 11.7 |

# H. Visualization of Growing graphs

This section helps us visualize the phases of our growing graphs. We have generated a synthetic graph of 60 nodes using PyGSP-Sensor and HE methods mentioned in Appendix D. We then added 40 new nodes denoted using black color in these existing graphs at four different timestamps. Figure 10 and Figure 11 shows the learned graph structure after each timestamp for two different Synthetic graphs.

**PyGsp**

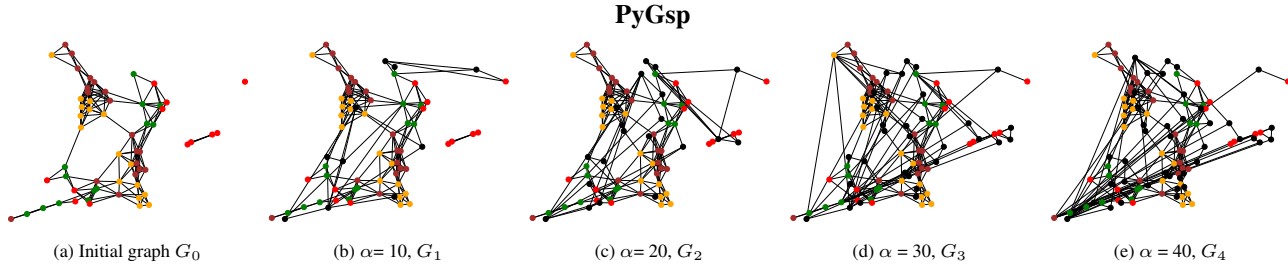

(a) Initial graph $G_0$     (b) $\alpha = 10$, $G_1$     (c) $\alpha = 20$, $G_2$     (d) $\alpha = 30$, $G_3$     (e) $\alpha = 40$, $G_4$

Figure 10: This figure illustrates the growing structure learned using GraphFlex for dynamic nodes. New nodes are denoted using black color, and $\alpha$ denotes number of new nodes. *PyGsp* denotes type synthetic graph.

**HE**

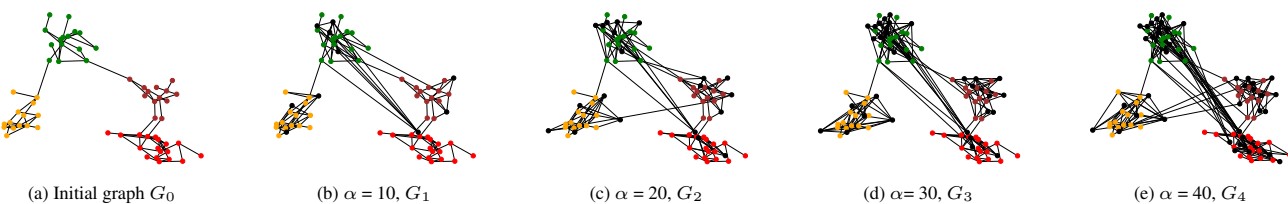

(a) Initial graph $G_0$     (b) $\alpha = 10$, $G_1$     (c) $\alpha = 20$, $G_2$     (d) $\alpha = 30$, $G_3$     (e) $\alpha = 40$, $G_4$

Figure 11: This figure illustrates the growing structure learned using GraphFlex for dynamic nodes. New nodes are denoted using black color, and $\alpha$ denotes the number of new nodes. *HE* denotes the type of synthetic graph.

**Vanilla**

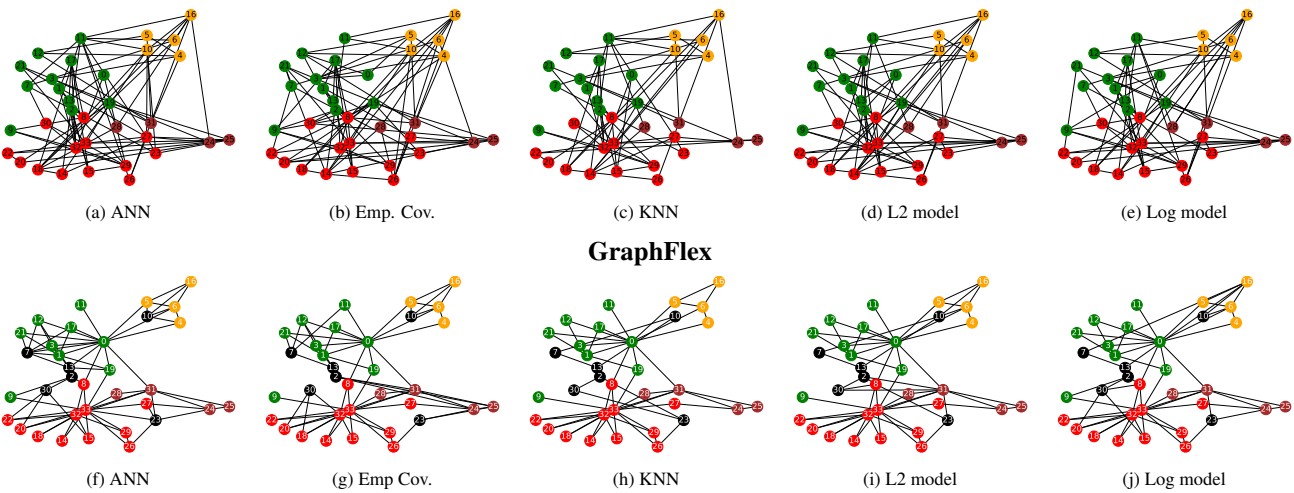

(a) ANN     (b) Emp. Cov.     (c) KNN     (d) L2 model     (e) Log model

**GraphFlex**

(f) ANN     (g) Emp Cov.     (h) KNN     (i) L2 model     (j) Log model

Figure 12: This figure compares the structures learned on Zachary's karate dataset when existing methods are employed with GraphFlex and when existing methods are used individually. We consider six nodes, denoted in black, as dynamic nodes.

## I. Structure Comparison on Karate Dataset

This section involves a comparison of the graph structure learned from GraphFlex with existing methods. Six nodes were randomly selected and considered as new nodes. Figure 12 visually depicts the structures learned using GraphFlex compared to other methods. It is evident from the figure that the structure known with GraphFlex closely resembles the original graph structure. Figure 13 shows the original structure of Zachary's karate club network (Zachary, 1977). We assumed six random nodes to be dynamic nodes, and the structure learned using GraphFlex compared to existing methods is shown in Figure 12.

## J. Clustering Quality

Figure 14 shows the PHATE (Moon et al., 2019) visualization of clusters learned using GraphFLEx's clustering module $\mathcal{M}_{clust}$ for $Xin$, $MNIST$, and $Baron - Human$ datasets.

*Original Karate Graph*

Figure 13: Original Karate Graph

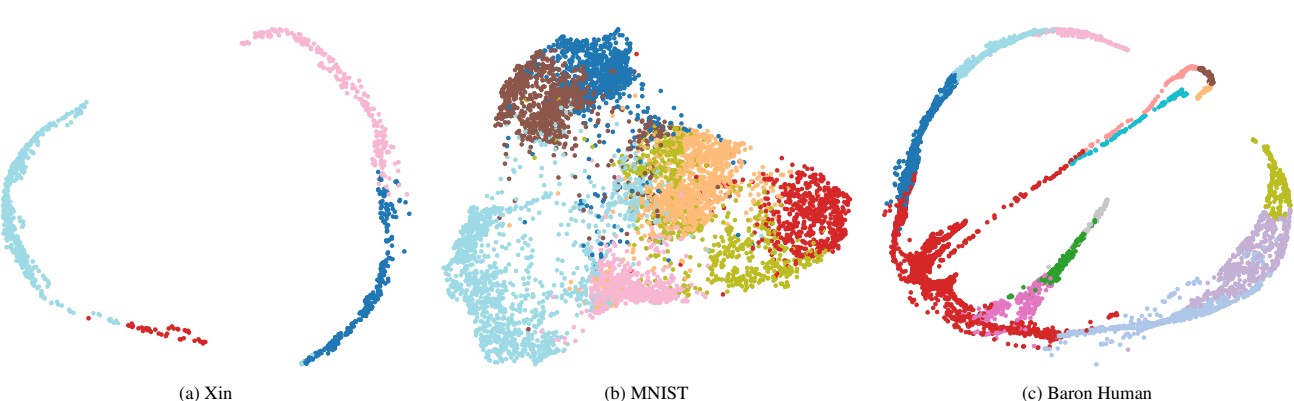

(a) Xin        (b) MNIST        (c) Baron Human

Figure 14: PHATE visualization of clusters learnt using GraphFlex clustering module for scRNA-seq datasets.

