# OpenReview forum: "GraphFLEx: Structure Learning $\underline{\text{F}}$ramework for $\underline{\text{L}}$arge $\underline{\text{Ex}}$panding $\underline{\text{Graph}}$s"
_ICML.cc/2025/Conference — Submitted to ICML 2025_

### Official Review · Reviewer_9WbL · 2025-03-14

**Overall Recommendation:** 4

**Summary:**

This paper proposes a graph structure learning framework GraphFLEx for large and expanding graphs, which consists of three modules: graph clustering, graph coarsening and graph learning. By leveraging clustering and coarsening, it improves the efficiency by restricting possible connection to only relevant nodes. Moreover, it also provides theoretical guarantees that the structure learned from the small subset of nodes is equivalent to that learned from the full set. Extensive experiments have been conducted to verify the effectiveness of the proposed method.

**Claims And Evidence:**

Yes. All the authors’ claims are supported by clear and convincing theoretical analysis and experiments.

**Essential References Not Discussed:**

No. All essential references are currently cited/discussed in the paper.

**Experimental Designs Or Analyses:**

Yes. I have checked the soundness of the experimental settings and results analysis. Extensive experiments have been conducted on 22 datasets, and the experimental designs and analyses look sound.

**Methods And Evaluation Criteria:**

Yes. The proposed method and evaluation criteria in the paper make sense for the problem and application.

**Other Comments Or Suggestions:**

I do not have any additional comments or suggestions for the authors.

**Other Strengths And Weaknesses:**

S1. The authors provide theoretical guarantees that the structure learned from the smaller graph coarsened from the communities of the input graph is equivalent to that learned from the original graph. This is also supported by the experimental results.

S2. The proposed framework, GraphFLEx, is flexible and supports 48 distinct methods for learning graph structures.

S3. GraphFLEx effectively controls computational growth, achieving near-linear scalability.

S4. Extensive experiments have been conducted and the results on four tasks demonstrate the effectiveness of the proposed method.

W1. The clustering and coarsening methods used in the framework, such as K-means and spectral clustering, are somewhat outdated and do not integrate or analyze the latest methods in the field.

W2. Downstream tasks, such as link prediction and graph classification, should be used to evaluate the proposed method.

**Questions For Authors:**

Q1. Since the methods in the framework are not the latest, how can newer methods be integrated with GraphFLEx and their effectiveness analyzed?

Q2. One of my concerns is how well GraphFLEx performs on downstream tasks, such as link prediction and graph classification, which are crucial for evaluating its effectiveness.

**Relation To Broader Scientific Literature:**

This paper proposes a graph structure learning framework based on clustering and coarsening techniques, achieving a 3x speedup compared with state-of-the-art methods in large graphs.

**Theoretical Claims:**

Yes. I have checked the correctness of theoretical claims in this paper.

---

> ### Author Rebuttal · Authors · 2025-03-31
>
> We sincerely thank the reviewer for their thoughtful feedback and for recognizing the potential of our work. We appreciate the reviewer’s positive response and for highlighting areas where the manuscript can be further improved.
>
> **W1 and Q1)** We would like to clarify that along with *K-means* and *spectral clustering*, *GraphFLEx* also employs some very recent and advanced methods for both clustering and coarsening. For clustering, we employ DMoN (JMLR 2023); for coarsening, we integrate FACH (2023), FGC (JMLR 2023), and UGC (NeurIPS 2024), all of which are recent and well-established techniques, ensuring GraphFLEx aligns with the latest advancements in graph learning.
>
> There are 3 different modules in GraphFLEx : clustering, coarsening, and the learning module. All 3 controls distinct properties. Altering any of these modules results in a new graph learning method. This flexibility allows seamless integration of newer methods, ensuring that GraphFLEx remains scalable and adaptable.
>
> **W2 and Q2)** We appreciate the reviewer’s concern and have included the link prediction experiments in the table below. As shown, the structure learned by GraphFLEx performs well on link prediction, sometimes even surpassing the results on the base structure, demonstrating its effectiveness. The table below follows the same configuration as presented in Table 4 in the main manuscript.
>
> | Dataset  | Base| (ANN vanilla) | (ANN Gflex) | (KNN vanilla) | (KNN Gflex) | (log vanilla) | (log Gflex) | (l2 vanilla) | (l2 Gflex)  | (covar vanilla) | (covar Gflex) | (large vanilla) | (large Gflex) |
> |----------|------------|---------------|------------|---------------|------------|---------------------|------------------|--------------------|----------------|----------------|--------------|------------------------|---------------------|
> | DBLP     | 95.13      | 96.57         | 96.61      | OOM           | 94.23      | OOT                 | **97.59**            | OOT                | **97.59**          | 97.22          | **97.59**       | OOT                    | 96.24               |
> | Citeseer | 90.78      | 80.12         | 96.32      | 85.17         | 96.24      | 80.48               | 96.24            | 80.48              | **96.48**          | 82.05          | 96.24       | 84.5                   | 94.38               |
> | Cora     | 89.53      | 84.47         | 95.3       | 79.23         | 95.14      | 90.63               | **95.45**            | 90.81              | 95.14          | 86.05          | 95.3        | 90.63                  | 94.67               |
> | Pubmed   | 94.64      | 94.24         | 96.91      | OOM           | **97.42**      | OOT                 | **97.42**            | OOT                | 97.37          | 94.89          | 94.64       | OOT                    | 94.41               |
> | CS       | 95         | 94.21         | 95.73      | OOM           | **96.02**      | OOT                 | 93.17            | OOT                | 93.17          | 93.52          | 92.31       | OOT                    | 95.73               |
> | physics  | 93.96      | **95.77**         | 91.34      | OOM           | 94.63      | OOT                 | 90.79            | OOT                | 94.63          | 92.03          | 90.79       | OOT                    | 92.97               |
>
> For graph classification, its applicability depends on the nature of the dataset. This task often involves multiple small subgraphs, as seen in applications like molecule or drug discovery, where subgraphs are inherently small. Since GraphFLEx integrates clustering, coarsening, and learning, the first two steps become redundant in such scenarios. Applying clustering and coarsening to small subgraphs may lead to unnecessary computational overhead without adding value.
>
> However, GraphFLEx can still be effectively applied to graph classification tasks by bypassing the clustering and coarsening steps and directly using its learning module to train models for classification. This flexibility ensures that GraphFLEx can adapt to different types of downstream tasks while maintaining its efficiency.
>
> **Appeal to Reviewer:** We again thank you for the insightful comments on our work. We will incorporate your suggestions into the revised manuscript. Please let us know if there are any remaining concerns or clarifications needed. As we approach the final stage, we would greatly value your positive support.
>
> Best regards,
>
> Authors

---

### Official Review · Reviewer_3rip · 2025-03-14

**Overall Recommendation:** 2

**Summary:**

This paper addresses graph structure learning, a critical challenge in graph machine learning. In contrast to standard strategies, the proposed solution, GraphFLEx, is particularly effective in large, expanding graph scenarios. By leveraging clustering and coarsening techniques, GraphFLEx significantly reduces computational costs while enhancing scalability. Notably, it supports 48 flexible methods, unifying clustering, coarsening, and graph learning into a comprehensive framework. Extensive experiments demonstrate the effectiveness of the proposed approach.

**Claims And Evidence:**

Yes

**Essential References Not Discussed:**

This paper may overlook some important references in the area of unsupervised graph structure learning. For instance, a set of leading graph structure learning methods are benchmarked in [1]. In particular, SUBLINE [2] is a state-of-the-art unsupervised graph structure learning method that merits discussion.

[1] OpenGSL: A comprehensive benchmark for graph structure learning, NeurIPS Track on Datasets and Benchmarks, 2023.

[2] Towards unsupervised deep graph structure learning, WWW, 2022.

**Experimental Designs Or Analyses:**

I checked all the experimental designs.

**Methods And Evaluation Criteria:**

Yes

**Other Comments Or Suggestions:**

No

**Other Strengths And Weaknesses:**

Pros.
- The paper is well-organized and easy to follow, featuring illustrative demonstrations.
- The idea of performing graph structure learning by integrating clustering and coarsening techniques is novel and intriguing.
- Extensive experiments on 22 different datasets validate the effectiveness of the proposed methods.

Cons.
- While the authors present theoretical justifications for combining clustering and coarsening for large, expanding graphs, the overall technical contributions appear limited.
- The framework unifies 48 methods via a pipeline of graph clustering, coarsening, and learning, which is compelling. However, offering practical insights into which method combinations work best would clarify the value of this flexibility.
- The term “large” in the title may be overstated. While scaling graph structure learning to millions or billions of nodes is challenging, evaluating GraphFLEx on larger benchmarks (e.g., ogbn-arxiv, ogbn-products) would strengthen its claims of scalability.
- The proposed method heavily depends on the quality of the underlying graph clustering. Since no single clustering algorithm consistently excels across diverse real-world graphs, this dependency could undermine the method’s effectiveness.

**Questions For Authors:**

Please see the comments above.

**Relation To Broader Scientific Literature:**

Graph structure learning is a pivotal research area within graph machine learning, positioning this paper within a broad and active scientific literature.

**Theoretical Claims:**

I checked the theoretical claims

---

> ### Author Rebuttal · Authors · 2025-03-31
>
> We thank the reviewer for the thoughtful feedback and for recognizing the potential of our work. We appreciate the detailed insights and suggestions for improvement.
>
> **Missing references:** Thankyou for pointing this out. We have added Table1 comparing vanilla SUBLINE[2] vs. GFLEx, where GFLEx significantly outperforms SUBLINE. We assure that pointed references will be added in the revised paper.
>
> Table1: Time(sec) and acc(%).
> |Data|Time-Van|Time-GFlex|Acc-Van|Acc-GFlex|
> |-|-|-|-|-|
> |Cite|8750|670|66.36|64.93|
> |Cora|7187|493|66.79|71.24|
> |DBLP|OOM|831|OOM|69.06|
> |Pub|OOM|914|OOM|70.94|
> |CS|OOM|1049|OOM|68.92|
>
> Also, any other SOTA methods can be easily integrated into GFLEx, as it scales structure learning using existing techniques. In terms of computational time, GFLEx remains the preferred choice since it learns the structure using only a concise set of potential nodes.
>
> **A1:** We acknowledge that GFLEx builds on existing methods to scale GSL. However, while much research has focused on improving graph model architectures, GSL itself remains underexplored. GFLEx addresses this gap by introducing a novel combination of clustering and coarsening to reduce the number of potential nodes for structure learning.
>
> Beyond efficiency GFLEx offers 48 structure learning options integrating clustering, coarsening, & learning into a single pipeline. This unified framework enables diverse structure learning strategies without multiple tools. We believe these contributions significantly advance structure learning especially for large, expanding graphs.
>
> **A2:** Thankyou for your thoughtful comment. Currently, GFLEx supports *K-means, Spectral & Deep Learning-based clustering methods*, each with unique strengths suited to different scenarios:
> * K-means is computationally efficient & works well when clusters have a well-defined spherical structure. It is useful for large-scale datasets where speed is a priority.
> * Spectral Clustering leverages eigenvalue decomposition making it effective for capturing complex graph structures, especially when communities are not clearly separable using simple distance metrics. However, it can be computationally expensive for large graphs.
> * Deep Learning-based Clustering adapts well to non-linear & high-dimensional patterns, making it a good choice for complex & feature-rich graph data, though it requires more computational resources.
>
> We will incorporate this discussion in Appendix
>
> **A3:** Thankyou for highlighting this. In response, we have included experiments on larger datasets: ogbn-arxiv (169K nodes), ogbn-products (2.45M), Flickr (89K), and Reddit (233K). Please note that given the large node count these experiments were conducted on a different machine distinct from the one used for earlier experiments.
> Spec.: Intel Xeon Platinum 8360Y CPU, 1.0 TiB RAM, & NVIDIA RTX A6000 (48 GiB VRAM).
>
> Table2: Time(sec)
> |Method|arxiv||products||flickr||reddit||
> |-|:-:|-|:-:|-|:-:|-|:-:|-|
> ||Van|GFlex|Van|GFlex|Van|GFlex|Van|GFlex|
> |Covar|OOM|3709|OOM|83145|2353|682|OOM|6676|
> |ANN|7836|4835|OOM|89312|2578|705|12679|6145|
> |knn|8318|6183|OOM|91860|2783|920|15609|6979|
> |l2|OOT|9012|OOT|OOT|93340|1292|OOT|5180|
> |log|OOT|45639|OOT|OOT|OOT|18752|OOT|60335|
> |large|OOT|5612|OOT|OOT|OOT|2289|OOT|9313|
>
> Table3: Node classification acc
> |Method|arxiv (60.13)||products (73.72)||flickr (44.92)||reddit (94.15)||
> |-|:-:|-|:-:|-|:-:|-|:-:|-|
> ||Van|GFlex|Van|GFlex|Van|GFlex|Van|GFlex|
> |Covar|OOM|60.26|OOM|68.23|44.65|44.34|OOM|94.13|
> |ANN|60.14|60.22|OOM|67.91|44.09|44.92|94.14|94.18|
> |knn|60.09|60.23|OOM|68.47|43.95|44.73|94.14|94.15|
> |l2|OOT|58.39|OOT|OOT|44.9|44.32|OOT|93.47|
> |log|OOT|58.72|OOT|OOT|OOT|44.59|OOT|94.13|
> |large|OOT|60.2|OOT|OOT|OOT|44.45|OOT|93.71|
>
> Table2 shows that GFLEx scales effectively to larger datasets & is the fastest among all baselines. Notably, methods like log, L2 & large models fail even on Flickr dataset, where GFLEx successfully scales them on Flickr, arxiv & Reddit. However, due to high computational cost of these methods GFLEx is unable to scale structure learning for them on ogbn-products dataset. We aim to further improve GFLEx's scalability in future work.
>
> Table 3 highlights the quality of learned structure on these large datasets through node classification acc. GFLEx maintains acc comparable to base structure (shown in parentheses with dataset).
>
> **A4:** To evaluate the effectiveness of these clustering methods we have conducted experiments (Table5) using NMI, Conductance & Modularity. Since clustering in GFLEx is applied only once on a randomly sampled small set of nodes selecting right method can be considered as part of hyperparameter tuning, where these clustering measures can guide the optimal choice based on dataset characteristics.
>
> **Appeal to Reviewer:** Thankyou again for your insightful comments. Please let us know if any concerns remain. Otherwise, we would really appreciate it if you could support the paper by increasing the score.
>
> Best regards,
>
> Authors

---

### Official Review · Reviewer_mkUX · 2025-03-16

**Overall Recommendation:** 2

**Summary:**

The article proposes a new framework for graph structure learning in large and expanding graphs. The key challenges addressed include the high computational costs and memory demands of existing methods, especially when dealing with dynamically growing graphs.

**Claims And Evidence:**

A formal complexity analysis showing under what conditions the framework scales linearly or sub-linearly is missing.
A breakdown of how each module contributes to computational cost is missing.

**Essential References Not Discussed:**

I cannot come up with any missing reference

**Experimental Designs Or Analyses:**

Yes, I checked the node classification and clustering quality results.

**Methods And Evaluation Criteria:**

yes, the criteria and methods are appropriate

**Other Comments Or Suggestions:**

- Fig 1: is the memory failure due to "fewer" than 10K nodes or more than 10K nodes?
- What is the definition of OOT or OOM for your resources?
- Table 3: The results are in seconds?

**Other Strengths And Weaknesses:**

The article addresses an interesting and valuable problem, but the motivation for the problem is introduced too late. As a result, I struggled to identify a clear set of strong points early on. Given the importance of the topic, I believe the article has significant potential for improvement in a future revision.

- The methodology needs to be more clearly defined and motivated. It was only around Section 3.1 that I fully understood the core issue being tackled.
- Assumptions, such as Assumption 1, lack sufficient justification, making it difficult to follow the argument.
- Figure 2 does not clearly illustrate how the proposed approach works or what the methodology entails. The methodology should be grounded in an established theoretical framework.

**Questions For Authors:**

- What are the main insights that led you to develop the methodology in this article?
- What are the SOTA results and their performance on your studied datasets?

**Relation To Broader Scientific Literature:**

The idea of using partial dynamic views on a graph to complete graph level tasks is a quite important problem, and this is quite valuable.

**Theoretical Claims:**

I checked the theorems in the appendix

---

> ### Author Rebuttal · Authors · 2025-03-31
>
> We thank the reviewer for the thoughtful feedback and for recognizing the potential of our work. We appreciate the detailed insights and suggestions for improvement.
>
> **Complexity**: We clarify that Sec3.6 and Table2 breaks down the complexity for both (a)best & (b)worst scenarios, highlighting each module. To improve, we explain each module's contribution and highlight GFLEx's linear & sub-linear time complexity. In clustering: kNN is the fastest $O(k^2)$, while spectral clustering is slowest $O(k^3)$. Since clustering is applied to randomly sampled, smaller subgraph with $k \ll N$ nodes, the cost is constant. For coarsening: FACH & UGC achieve best complexity: $O(\frac{k_\tau}{c})$, while FGC denotes the worst case with $O((\frac{k_\tau}{c})^2 |S_\tau^i|)$, where $c$ is number of communities, $|S_\tau^i|$ is number of coarsened nodes & $k_\tau$ is number of nodes at time $\tau$. For learning module, ANN is the most efficient with $O(N \log N)$, while GLasso is worst with $O(N^3)$. Therefore, GFLEx’s overall complexity is bounded between $O(k^3 + (\frac{k_\tau}{c})^2 |S_\tau^i| + \alpha^3)$ and $O(k^2 + \frac{k_\tau}{c} + \alpha \log \alpha)$, where $\alpha = |S_\tau^i| + |E^i_\tau|$. $M_{clust}$ is trained once keeping its runtime bounded, $M_{coar}$ also remains controlled as some methods have linear complexity. Thus, both of these contribute linearly to the overall time, denoted as $O(N)$. The total complexity of GFLEx is $O(N + M_{gl}(|S_i, X^i\tau|))$, **scaling linearly or sub-linearly** based on $\alpha$ & $M_{gl}$. For eg, ANN maintains linear complexity if $\alpha \log(\alpha) \approx N$, while GLasso exhibits linear behavior when $\alpha^3 \approx N$.
>
> **Motivation:** The Intro is structured to gradually build motivation, first highlighting the ubiquity of graph data (Intro:para1) & the necessity of GSL (para2). Para3 states key challenges:scalability & adaptability before presenting GFLEx(para4). Sec2 formally defines these challenges (Goal1&2), & Sec3 details how GFLEx efficiently achieves these goals. To enhance we will refine Intro by adding:
>
> "Real-world graphs continuously expand for eg, e-commerce networks accumulate new transactions daily, academic networks grow with new publications, financial & social graphs evolve with ongoing interactions. These dynamic changes require efficient methods that can incrementally learn graph structures rather than recomputing them from scratch. However, existing approaches struggle with expanding graphs."
>
> This will provide a stronger motivation by first illustrating real-world scenarios where graph expansion is inevitable then naturally transitioning into limitations of SOTA.
>
> **Methodology, Figure & Theoritical Framework:** Our intention was to introduce the methodology progressively across multiple sections to maintain a coherent narrative. To enhance understanding, we’ve updated Fig2 & refined caption summarizing the methodology, **updated Fig2: https://t.ly/URWmZ**.
>
> Additionally Theorem1 states that with a constant probability of success the neighbor of incoming nodes $N_k(E_i)$ can be effectively recovered using GFLEx’s multi-step approach to establish the theoretical foundation.
>
> **Assumption1:** It is grounded in the well-established homophily principle which forms the basis of most graph coarsening & learning methods. To formalize this we assume that the generated graphs follow the DCSBM, an extension of SBM that accounts for degree heterogeneity, making it a more flexible & realistic choice for real-world networks. We acknowledge that the justification for this assumption could be more explicit & we will enhance this explanation in the revised version to improve clarity.
>
> **Ans-Other Comments:**
> * We have revised it to: *Vanilla KNN failed to construct graph structures for more than 10K nodes due to memory limitations.*
> * Specifications used for experiments (also in Sec4): Intel Xeon W-295 CPU & 64GB RAM using Python environment; OOM: Execution failure due to memory constraints; OOT: Execution exceeding 100k seconds (~28 hours).
> * Yes results are in seconds.
>
> **Ans-Questions:**
> * Most graph research focuses on developing deep learning architectures, often overlooking the critical role of graph structure. Structure learning remains underexplored, with existing methods struggling to scale for large & expanding graphs. We aim to bridge this gap by using clustering & coarsening techniques to enable efficient structure learning at scale.
>
> * Table4 shows results of various GNN models on base structure reflecting SOTA performance. These results show that GFLEx maintains good node classification acc while outperforming existing structure-learning methods.
>
> **Appeal to Reviewer:** Thankyou again for your insightful comments. We will incorporate your suggestions into the revised manuscript. Please let us know if any concerns remain. Otherwise, we would really appreciate it if you could support the paper by increasing the score.
>
> Best regards
>
> Authors

---

### Decision · Program_Chairs · 2025-05-01

**Decision:**

Reject

**Comment:**

This paper introduces a flexible framework for structure learning on large and growing graphs, demonstrating practical benefits in efficiency and scalability. The design facilitates various combinations of clustering, coarsening, and learning methods. The authors support their claims with extensive experiments.
While the work shows promise, certain concerns remain, such as the lack of clear explanations for the selection of specific modules and whether the claim of handling "large" graphs is fully substantiated. The authors have provided thoughtful responses and additional experiments in the rebuttal, addressing many of these points.
Overall, the paper presents a valuable approach to scalable graph structure learning. However, further clarification and strengthening of the methodology would enhance its contribution.